# SGAR: Structural Generative Augmentation for 3D Human Motion Retrieval

**Jiahang Zhang    Lilang Lin    Shuai Yang    Jiaying Liu***
Wangxuan Institute of Computer Technology, Peking University
{zjh2020, linlilang, williamyang, liujiaying}@pku.edu.cn

## Abstract

3D human motion-text retrieval is essential for accurate motion understanding, targeted at cross-modal alignment learning. Existing methods typically align the global motion-text concepts directly, suffering from sub-optimal generalization due to the uncertainty of correspondence learning between multiple motion concepts coupled in a single motion/text sequence. Therefore, we study the explicit fine-grained concept decomposition for alignment learning and present a novel framework, **S**tructural **G**enerative **A**ugmentation for 3D Human Motion **R**etrieval (**SGAR**), to enable generation-augmented retrieval. Specifically, relying on the strong priors of existing large language model (LLM) assets, we effectively decompose human motions structurally into subtler semantic units, *i.e.*, body parts, for fine-grained motion modeling. Based on this, we develop part-mixture learning to better decouple the local motion concept learning, boosting part-level alignment. Moreover, a directional relation alignment strategy exploiting the correspondence between full-body and part motions is incorporated to regularize feature manifold for better consistency. Extensive experiments on three benchmarks, including motion-text retrieval as well as recognition and generation applications, demonstrate the superior performance and promising transferability of our method. Our project page can be found at https://jhang2020.github.io/Projects/SGAR/SGAR.html

## 1 Introduction

3D human motion understanding is a crucial topic in computer vision. To deal with the complex nature of human motions, natural language is widely adopted as a medium to achieve fine-grained motion modeling. Among these, motion-text retrieval is a fundamental task, which aims to search relevant motion/text samples based on queries of another modality, with wide applications including film production [22], health-care [43], and character animation [1].

Following the practice of the image-text [28, 3] and video-text [41, 49] retrieval, many efforts have been made to establish a semantic-aligned latent space of 3D motions and texts. TMR [25] first proposes the motion-text retrieval benchmark and presents a framework by jointly contrasting and reconstructing. However, the scarcity of high-quality motion-text data heavily limits the model performance. To this end, many works propose to leverage the pre-trained vision/language models as priors to boost the motion-text retrieval. For instance, some methods [35, 44] render the 3D motions into images/videos to leverage large-scale pre-trained vision-language models, *e.g.*, CLIP [28], as priors to bridge the motion and text modalities.

However, we argue that the above approaches can still suffer from two points. First, the significant disparity between rendered and natural images results in sub-optimal representations for the vision-bridged methods. More importantly, they neglect the characteristics of human motion data. For

---

*Corresponding Author.

39th Conference on Neural Information Processing Systems (NeurIPS 2025).

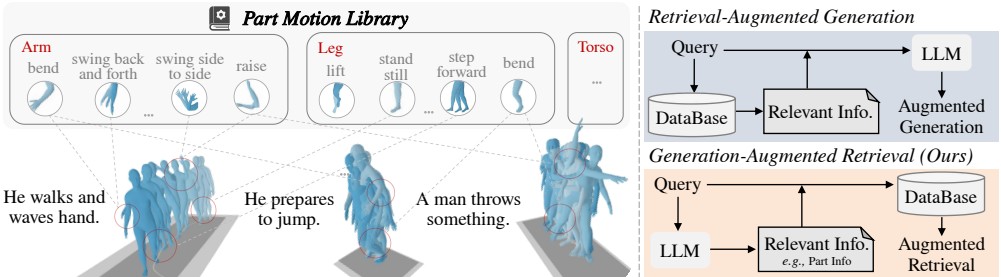

Figure 1: *Left*: A motion sequence typically comprises multiple motion concepts, *e.g.*, body part movements. Meanwhile, these fine-grained concepts often exhibit higher cross-motion generality, facilitating more precise and generalized motion-text alignment. *Right*: Inspired by the *Retrieval Augmented Generation* [7] (RAG), we explore the **Generation-Augmented Retrieval** paradigm, which relies on LLMs as available assets, and propose a series of effective alignment strategies.

instance, the subtler semantic units, *i.e.*, body parts that depict motions in a finer granularity are ignored, leading to sub-optimal alignment. Taking "walks and waves hand" shown in Figure 1 as an example, with a single global alignment objective, existing methods may incorrectly correspond the motion pattern of arm swinging to the verb "walk" rather than "wave hand", especially training with limited data scale. This means the difficulty of motion-text retrieval arises not only from the cross-modal alignment but also from the reorganization and decomposition of semantic concepts within the same modality. In this context, learning the correspondence of local concepts as primitives can lead to more precise, generalized local alignment, and ultimately enhance global understanding. Therefore, our key motivation is to explicitly decompose the holistic human motions into fine-grained local concepts by leveraging available priors, *e.g.*, LLMs, to achieve better motion-text alignment.

To this end, we study a new *Generation-Augmented Retrieval* (GAR) paradigm as shown in Figure 1, which is less explored by previous work, *i.e.*, leveraging the generated information (part motion texts) relevant to the queries (full motion text) to augment retrieval performance. Specifically, to exploit the generated structural knowledge, we propose a novel Structural Generative Augmentation Retrieval framework, *SGAR*. SGAR aligns linguistic motion knowledge of body parts generated by LLMs for fine-grained concept modeling. Based on this, a *part-mixture learning strategy* is developed to better de-correlate the motion pattern learning of different parts and the full-body. By virtue of such a way, the consistency learning at both full-body and part levels is boosted. In addition to this independent intra-hierarchy alignment, we propose a *directional consistency regularization*, introducing relational knowledge between full-body and part motions to enhance correspondence. Finally, we conduct extensive experiments on three large-scale benchmarks to demonstrate our strong performance on motion-text retrieval and promising transfer capacity to a series of downstream tasks.

Our contributions can be summarized as follows:

- We propose *SGAR*, an effective motion-text retrieval framework with structural generative knowledge augmentation. By modeling part motion concept correspondence as more general knowledge, it effectively boosts the cross-modal alignment performance of motion-language models.

- To enable part-based motion alignment, we integrate structural linguistic knowledge from LLMs and propose a part-mixture learning strategy. By decoupling the motion representations of the global body and local parts, our method can facilitate alignment within different structural levels.

- In addition to independently minimizing the Euclidean distance of embeddings at global and local motion levels, a directional alignment objective is introduced to model the relational knowledge between them. This further alleviates over-fitting and leads to better representation consistency.

## 2 Related Works

### 2.1 Motion-Language Retrieval and Alignment Learning

Cross-modal representation alignment enables entities from different modalities, which share conceptually similar meanings, to have comparable feature embeddings. However, different from images,

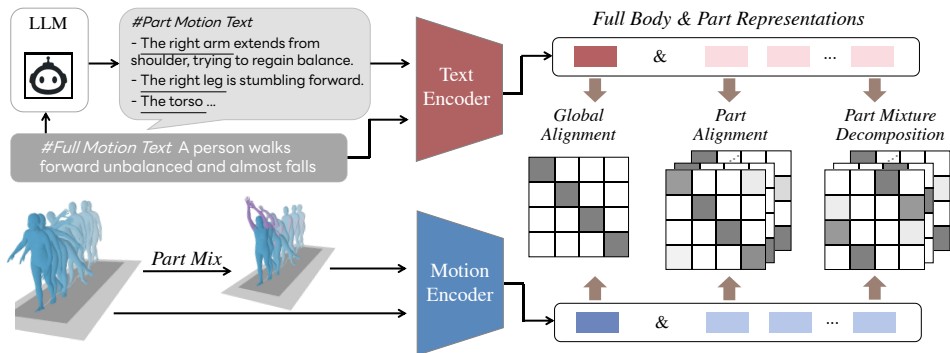

Figure 2: Overview of *SGAR* pipeline following a GAR paradigm. We utilize LLMs to generate the part motion corpus and construct mixed motion data, to enable the part motion modeling. Three objectives are integrated: 1) global alignment for full body motion-text pairs, 2) part alignment with soft contrastive learning; 3) part-mixture decomposition for further de-correlation of different part motions. For clarity, the directional relation regularization is omitted here.

human motion-text data collection presents more challenges, *e.g.*, the diverse semantics and the limited motion-text data scale. Most existing approaches focus on incorporating pre-trained vision/language models to guide motion representation alignment. For instance, TMR [25] directly utilizes a frozen pre-trained text encoder [32] to accelerate learning of the motion encoder. MotionCLIP [35] and TriModal [44] convert motion data into images or videos, enabling the use of pre-trained vision-language models for guidance. Yu *et al*. [45] introduced a novel motion representation that structures 3D motion data into patches, making it compatible with pre-trained ViT [4] models. Other studies [33, 42] take a model-centric approach, aiming to achieve more informative motion representations. However, these methods overlook the inherent structural properties of human motion, which can lead to suboptimal representations and hinder generalization performance.

Meanwhile, some recent methods [11, 39] also study the part kinematics to boost motion generation performance. However, this cross-modal alignment is unidirectional, *i.e.*, the text encoder is often fixed, while the model is dominated by generation objectives, making it difficult to achieve explicit semantic alignment. Furthermore, the part knowledge is injected by guiding motion generation through conditioning, which still differs from our case, that is, achieving bidirectional fine-grained motion-language alignment.

## 2.2 Collaboration between Generation and Retrieval

Retrieval and generation are closely connected not only because of their similar function, *i.e.*, new content presentation, but also the potential collaboration. Retrieval-Augmented Generation (RAG) [7, 10] is widely studied to enhance accuracy and credibility of LLM generation by knowledge from external databases. In contrast, with more and more impressive LLMs as available knowledge expert assets, we explore a new *Generation-Augmented Retrieval* (GAR) paradigm, especially to leverage the strong priors in LLMs to augment data-scarce 3D human motion retrieval, which has been studied by few works. Mao [21] discussed generation-augmented retrieval to deal with language question answering (QA), where, however, "retrieval" is still evaluated by and serves for the generative QA task. Therefore, it is still confined to RAG in a sense. In contrast, we focus on the retrieval task exactly and aims to boost it by the generation knowledge following the GAR paradigm. Meanwhile, our cross-modal setting also brings new challenges in how to effectively utilize the generated information.

## 3 The Proposed Method

Our goal is to achieve semantic-aligned representations with paired motion-language dataset $\mathcal{D} = \{m_i, t_i\}$ to support cross-modal retrieval, where $(m_i, t_i)$ are the $i_{th}$ paired motion and text data.

## 3.1 Contrastive Motion-Language Learning

**Data Representation.** The human motions can be generally formulated as skeleton joint positions defined in SMPL [18] model, *i.e.*, $m_i \in \mathcal{R}^{T \times J \times 3}$, where $T$ is the sequence length and $J \times 3$ is the 3D positions of $J$ joints in Cartesian coordinates. Following previous work, we further structure the motion data into a series of "patches" to leverage the pre-trained ViTs as priors. Typically, a patch aggregates the joint information of a body part in some adjacent frames. The final input motion can be formulated as $m_i \in \mathcal{R}^{T \times PN \times 3}$ for patchification, where $P = 5$ is the body part number, *i.e.*, {*left/right arm, left/right leg, torso*}, and $N$ is the patch size. More details can be found in [45].

**Model Architecture.** Following the canonical design, our model consists of two modality-specific encoders, $f_m(\cdot)$ and $f_t(\cdot)$. Specifically, for the *motion encoder*, we adopt ViT-B/16 [4] with 12 layers pre-trained on ImageNet-21k [31] to better utilize the existing prior vision knowledge. For the *text encoder*, we employ the pre-trained Distill-BERT [32].

To obtain the motion/text embeddings, we append a $[cls]$ token to inputs, which is finally projected into the shared motion-language latent space, *i.e.*, $z_{m,i}, z_{t,i}$ for contrastive learning. The contrastive learning objective [23] can be formulated as the sum of the following two terms, denoted as $\mathcal{L}_{full}$:

$$\mathcal{L}_{full}^{m2t} = \mathbb{E}\left[-\log \frac{\exp(z_{m,i} \cdot z_{t,i}/\tau)}{\sum_j \exp(z_{m,i} \cdot z_{t,j}/\tau)}\right], \mathcal{L}_{full}^{t2m} = \mathbb{E}\left[-\log \frac{\exp(z_{m,i} \cdot z_{t,i}/\tau)}{\sum_j \exp(z_{m,j} \cdot z_{t,i}/\tau)}\right], \quad (1)$$

where $\tau$ is the temperature coefficient. For clarity, only text-to-motion ($t2m$) term is presented in the following, which shares similar definition of motion-to-text ($m2t$) term.

## 3.2 Structural Knowledge Generation

With the rapid development of LLMs, it is promising to leverage LLMs as available knowledge experts to boost the data-scarce motion-text retrieval scenario as discussed in Sec. 1. Specifically, we aim to generate the motion descriptions for each body part based on $t_i$.

***Remark***: Leveraging LLMs to generate part texts is also studied in other motion generation and recognition works [48, 40, 5]. Differently, we focus on the retrieval task following GAR paradigm, especially jointly considering the testing GAR application. Specifically, since there are often no paired motion-text data for the query side or database during testing, only texts can be conveniently utilized. It means some complex prompting strategies, *e.g.*, feeding LLMs with both motion pose descriptors and texts, are inapplicable. While there are inevitably some less accurate descriptions generated by LLMs, especially for complex motions, the output content is still relevant or partially matched to serve as *supplementary* and boost the retrieval learning. More discussions and analysis on the part texts generated by LLMs and model robustness can be found in Appendix A.1 and A.2.

To this end, we develop a clear and effective generation strategy. We select LLaMA3-70B [37] with carefully crafted few-shot prompts for our implementation. Meanwhile, to improve part text quality for training, we provide LLMs with all global texts annotated by different subjects as detailed and comprehensive descriptions to jointly reason (for training data generation only). Besides, to deal with the uncertainty in generation for the parts with confusing or hard-to-infer motions, we introduce a special indicator, $[udf]$, and require the LLMs to output this indicator for the parts with high uncertainty. Subsequently, the texts with this indicator are excluded during training. Overall, our GAR paradigm is compatible with different LLMs and generation strategies, where the retrieval performance can be expected to be boosted as the generation capability improves.

Then, the generated part motion descriptions are fed into the text encoder, yielding the embeddings $z_{t,i}^j$ for the $j_{th}$ body part text of the $i_{th}$ sample. On the motion encoder side, we register an additional token for each body part, *i.e.*, $[cls - p_j]$, to feed the encoder to obtain part motion embeddings $z_{m,i}^j, j = 1, ..., P$, denoting the $j_{th}$ part. Along with the $[cls]$ tokens, the motion encoder aligns both the global and local motion embeddings. However, the local part motions are much less diverse, and different body motions can share similar part motions. Therefore, we adopt a soft contrastive learning objective by replacing the hard label with the intra-modality similarity scores:

$$\mathcal{L}_{part}^{t2m} = \mathbb{E}\left[-\frac{1}{P}\sum_{j=1}^{P} p^{\tau'}\left(z_{t,i}^j, z_{t,*}^j\right)\log p^{\tau}\left(z_{m,i}^j, z_{t,*}^j\right)\right], p_k^{\tau}\left(z, z_{t,*}^j\right) = \frac{\exp(z \cdot z_{t,k}^j/\tau)}{\sum\limits_{i=1}\exp(z \cdot z_{t,i}^j/\tau)}. \quad (2)$$

### 3.3 Part-Mixture Decorrelation Learning

Due to the inherent correlation between global and part motions, it can be undesirably learned by model to derive the part embeddings based on the common concurrence of motion patterns, instead of the exact part kinematics. For instance, the model can associate arm swinging with leg stepping, which can impair knowledge generalization. Therefore, we propose a part-mixture learning policy to construct unusual motion samples to force the representation decorrelation and precise modeling.

With the part "patches" representation, we can simply perform Cut-Mix [46] by exchanging the part patches of different motions. Specifically, given two motion data $m_i, m_k$, we randomly sample $K$ body parts ($K < P$) and replace the corresponding patches of $m_i$ with $m_k$ to generate the mixing motion data. Then we obtain the full body and part motion embeddings as Sec. 3.2 by feeding the mixed motions into the motion encoder.

Similarly, we optimize the global and part alignment objectives. For global alignment, we utilize the linear blending of text embeddings, $z_{t,i}$ and $z_{t,k}$, as the target text positives for contrastive learning. Specifically, it is calculated as $z_{t,mix} = \lambda_m z_{t,i} + (1 - \lambda_m) z_{t,k}$, where $\lambda_m$ is the mixing ratio determined by the proportion of replaced part number. For part alignment, the target part text embeddings of mixed data can be reassembled from $z_{t,i}^j$ and $z_{t,k}^j$ following the replacement in mixing process. Building on this, we calculate the $\mathcal{L}_{full}^{mix}$ and $\mathcal{L}_{part}^{mix}$ for optimization similar to Eq. (1) and Eq. (2) to boost the part-mixture learning. More detailed formulations can be found in Appendix D.1.

***Remark***: Although this direct copy-paste approach can lead to globally unreal motions, recent work [16] has emphasized the *local motion patterns* as a more essential aspect for human motion modeling. Our experiments in Table 2 also verify the effectiveness of this copy-paste implementation.

### 3.4 Directional Relation Alignment Strategy

As introduced before, we encourage the model to jointly learn both full-body and part alignment. However, the two objectives optimize the Euclidean distance of embeddings independently, which ignores the relational knowledge between them, leading to over-fitting and limited performance. Inspired by the inter-word directional knowledge discovered in the neighbor topology structure of language model latent space [38], we explore the motion-text analogies between the relevant global-local motion concepts as a beneficial representation regularization.

Specifically, we model the relational knowledge between the full body and parts as the direction of their difference in the normalized space. Formally, the directional relation alignment objective can be denoted as

$$\mathcal{L}_r = \mathbb{E}\left[1 - \frac{1}{P}\sum_{j=1}^{P} \frac{\Delta z_{m,i}^j \cdot \Delta z_{t,i}^j}{|\Delta z_{m,i}^j||\Delta z_{t,i}^j|}\right], \tag{3}$$

where $\Delta z_{m,i}^j = z_{m,i} - z_{m,i}^j, \Delta z_{t,i}^j = z_{t,i} - z_{t,i}^j$. This can be regarded as a part-conditioned motion knowledge consistency regularization, leading to more meaningful representations. Overall, the full optimization objective of our framework can be formulated as ($\lambda_*$ is the loss weight)

$$\mathcal{L} = \mathcal{L}_{full} + \mathcal{L}_{part} + \lambda_{mix}\mathcal{L}_{full}^{mix} + \lambda_{mix}\mathcal{L}_{part}^{mix} + \lambda_r\mathcal{L}_r. \tag{4}$$

### 3.5 Generation-Augmented Retrieval: An Optional Testing-Augment Strategy

The above designs naturally enable a testing-augmentation option by providing additional part information. Specifically, giving one or more additional part texts embeddings, we calculate the similarity matrixes between both global and local motion-text pairs, which are then summed with a weight $\alpha$, *i.e.*, $s = s_g + \alpha \sum_{j \in J_p} s_p^j$, where $s_g, s_p^j$ and $s$ are the global, part, and final similarity matrix and $J_p$ is the set of parts with available descriptions. Finally, the retrieval results can be derived from $s$. This is an *optional* generation-augmented retrieval policy during testing. In implementation, $\alpha$ is set as 0.1 to control the intensity of the part-level effect.

Next, some possible application scenarios are discussed. For text-to-motion retrieval, part texts can be provided by users as additional queries or generated by LLMs. For motion-to-text retrieval, the additional part texts can be automatically generated by feeding global texts in the database to LLMs and maintained in the database.

Table 1: Motion-text retrieval results on HML3D and KIT benchmark. "SGAR" follows the same test setting as other methods (*i.e.*, no part information available for testing), while "SGAR++" employs part information during testing as discussed in Sec. 3.5 to show an optional *generation augmented retrieval* capacity.

| Protocol | Methods | Text-to-motion retrieval | | | | | Motion-to-text retrieval | | | | |
|---|---|---|---|---|---|---|---|---|---|---|---|
| | | R@1↑ | R@3↑ | R@5↑ | R@10↑ | MedR↓ | R@1↑ | R@3↑ | R@5↑ | R@10↑ | MedR↓ |
| All-HML3D | TMR [25] | 8.92 | 16.33 | 22.06 | 33.37 | 25.00 | 9.44 | 16.90 | 22.92 | 32.21 | 26.00 |
| | MoPatch [45] | 10.80 | 20.00 | 26.72 | 38.02 | 19.00 | 11.25 | 19.98 | 26.86 | 37.40 | 20.50 |
| | ChronRet [6] | 8.30 | 16.70 | 23.65 | 35.31 | 22.00 | 9.08 | 11.22 | 23.68 | 34.31 | 23.00 |
| | **SGAR** | **12.86** | **23.52** | **30.75** | **43.00** | **15.00** | **13.82** | **23.38** | **30.09** | **41.83** | **16.00** |
| | SGAR++ | 14.07 | 25.32 | 33.03 | 44.78 | 13.00 | 14.64 | 25.52 | 32.55 | 44.50 | 14.00 |
| Small-HML3D | TMR [25] | 67.45 | 86.22 | 91.56 | 95.46 | 1.03 | 68.59 | 86.75 | 91.10 | 95.39 | 1.02 |
| | MoPatch [45] | 71.61 | 90.02 | 94.35 | 97.69 | 1.00 | 72.11 | 90.21 | 94.44 | 97.76 | 1.00 |
| | TriModal [44] | 68.58 | 85.02 | 88.77 | 92.58 | 1.01 | 68.64 | 85.52 | 88.76 | 92.82 | 1.01 |
| | ChronRet [6] | 70.10 | 87.07 | 91.13 | 94.32 | 1.01 | 70.19 | 86.66 | 90.69 | 94.23 | 1.01 |
| | **SGAR** | **75.66** | **92.11** | **95.55** | **98.06** | **1.00** | **76.35** | **92.38** | **95.69** | **98.02** | **1.00** |
| | SGAR++ | 76.51 | 92.77 | 95.87 | 98.27 | 1.00 | 77.24 | 92.79 | 96.01 | 98.20 | 1.00 |
| All-KIT | TEMOS [24] | 7.11 | 17.59 | 24.10 | 35.66 | 24.00 | 11.69 | 20.12 | 26.63 | 36.39 | 26.50 |
| | TMR [25] | 10.05 | 20.74 | 30.03 | 44.66 | 14.00 | 11.83 | 22.14 | 29.39 | 38.55 | 16.00 |
| | MoPatch [45] | 14.02 | 28.91 | 34.10 | 50.00 | 10.50 | 13.61 | **27.54** | 33.33 | 44.77 | 13.00 |
| | **SGAR** | **16.75** | **30.48** | **40.00** | **53.61** | **9.00** | **14.46** | 25.54 | **35.54** | **48.31** | **11.00** |
| | SGAR++ | 16.51 | 30.96 | 40.48 | 56.99 | 8.00 | 16.14 | 29.28 | 38.67 | 52.29 | 9.00 |
| Small-KIT | TMR [25] | 50.00 | 78.02 | 87.97 | 94.87 | 1.50 | 51.21 | 78.64 | 89.00 | 95.31 | 1.50 |
| | MoPatch [45] | 53.55 | 79.82 | 88.92 | 96.29 | 1.36 | 54.54 | 79.68 | 89.35 | 96.11 | 1.31 |
| | TriModal [44] | 58.10 | 86.34 | 93.08 | 96.47 | 1.08 | 60.23 | 86.44 | 93.22 | 95.87 | 1.20 |
| | LaMP [14] | 52.50 | 84.70 | 92.70 | 97.60 | - | 54.00 | 84.40 | 92.20 | 97.60 | - |
| | **SGAR** | **62.12** | **87.88** | **94.25** | **98.00** | **1.04** | **63.00** | **87.25** | **94.38** | **98.38** | **1.03** |
| | SGAR++ | 62.50 | 88.12 | 94.75 | 98.75 | 1.04 | 63.38 | 87.75 | 95.00 | 98.88 | 1.02 |

# 4 Experiments on Motion-Text Retrieval

## 4.1 Datasets and Implementation Details

**HumanML3D** [8] (HML3D) enriches the AMASS [20] and HumanAct12 [9] motions by with the textual descriptions. It consists of 23,384 and 4,380 motions for training and testing with a mirror augmentation, respectively. There are 3.0 different textual labels on average for each motion.

**KIT-ML** [26] (KIT) is a small dataset with a focus on locomotion motions, consisting of 3,911 sequences and 6,278 text descriptions. We obtain 4,888, 300, and 830 motions for training, validation and testing , respectively, each of which is annotated 2.1 times on average.

**Motion-X** [15] is a large-scale dataset with more than 80K sequences from different motion domains. The texts are generated by LLMs. We further process the motions into our adopted representation. Since there are few previous works evaluated on this dataset, it is mainly utilized to support cross-dataset evaluation of transfer capacity.

For implementation, we utilize the same data pre-processing methods with ViT-B and Distill-BERT as motion and text encoders following previous works [25, 45]. The model is trained using Adam optimizer [12] for 50 epochs, with learning rates of $10^{-5}$, $10^{-4}$ and $10^{-3}$ for the text encoder, the motion encoder, and the projection heads, respectively. The embedding dimension after projection is 256 for contrastive learning. $\lambda_{mix}$ and $\lambda_r$ are set to 0.5 and 0.1. The temperature coefficients $\tau$ and $\tau'$ are 0.07 and 0.05. The batch size is 128. For the input data, the patch size $N$ is 224. We pad or crop the motions to a fixed length of 224 following ViT. For evaluation, we adopt the metrics and protocol in [25]. Specifically, the standard Recall at various ranks, *i.e.*, R@k, and median rank, MedR, are employed to evaluate the performance. Meanwhile, we vary the scale and data similarity of the target gallery sets, yielding **All Data**[2], **Small** protocols following [25, 45]. More details in Appendix D.2.

## 4.2 Performance Comparison

In this section, we compare our method with the latest works with experimental settings aligned. As shown in Table 1, our method achieves the best scores with significant performance improvement

---

[2]Following [45], we consider misjudgments due to mirror augmentation for evaluation correction.

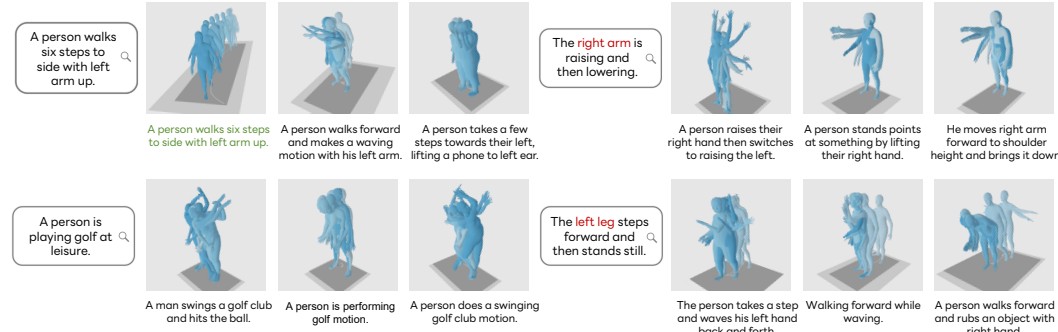

Figure 3: Qualitative T2M retrieval with top-3 results on HML3D testing set. The left column is *full-body retrieval*, with test set query (the first sample) and the customized text query (second) Green text is the perfect match. The right column is *part-based retrieval*, using only part motion queries.

.

Table 2: Ablation studies on our designs, part contrastive learning (Part CL), part-mixture learning (Part Mix), and directional relation regularization (Relation Reg.)

| Part CL | Part Mix | Relation Reg. | T-to-M Retrieval | | | M-to-T Retrieval | | |
|---|---|---|---|---|---|---|---|---|
| | | | R@1 | R@5 | MedR | R@1 | R@5 | MedR |
| | | | 9.87 | 26.36 | 19.50 | 10.19 | 26.56 | 21.50 |
| ✓ | | | 11.27 | 28.44 | 16.50 | 12.57 | 27.87 | 19.00 |
| ✓ | | ✓ | 11.90 | 28.56 | 16.00 | 12.95 | 28.47 | 18.00 |
| ✓ | ✓ | | 12.09 | 30.48 | 15.50 | 13.41 | **30.13** | 17.00 |
| ✓ | ✓ | ✓ | **12.86** | **30.75** | **15.00** | **13.82** | 30.09 | **16.00** |

compared with the state-of-the-art methods on both HML3D and KIT benchmarks. Previous works mainly focus on how to enable the availability of pre-trained vision/language models while ignoring the structural nature of human motion data. In contrast, our method starts from the characteristics of the human body and models the part kinematics in a GAR manner, achieving remarkable performance improvement and renewing the scores. Meanwhile, compared with Lex [19] which requires 4 training stages, our method is more training-friendly (∼ *8 times faster*) and clear with better performance.

Besides, benefited from our GAR paradigm, SGAR can be boosted with further generated (or user-provided) part information as discussed in Sec. 3.5, yielding a testing-augment version denoted as "SGAR++". *Although it can be unfair to directly compare SGAR++ with previous works without part information in testing, we show this intended to verify our motivation and demonstrate a remarkable testing-augment capacity.* As we can see Table 1, additional part information as supplementary can significantly benefit retrieval performance. More discussions and visualizations on this part-based testing GAR can be found in Appendix B.2 and C.1.

### 4.3 Qualitative Results

We present the text-to-motion retrieval results with *full-body and part motion text queries*, respectively, in Figure 3. For the full-body retrieval, our model successfully retrieves the ground-truth/semantic-aligned motion samples, whether by the original text annotations or the user-customized text queries. For the part-based retrieval, we obtain the candidate motions by calculating the similarity of their part motion embeddings and the given part texts, achieving the *partial match retrieval* [29]. Meanwhile, these two techniques can be combined to achieve more fine-grained motion retrieval. As we can see, although the results possess different global motion concepts, the motions of the target local regions align with the part text descriptions well, demonstrating our superior structure-aware modeling capacity. More visualization results can be found in Appendix B.

### 4.4 Ablation Study

In this section, we present the ablation study results on the HumanML3D dataset.

Table 3: Comparison for the different LLMs for part motion knowledge generation.

| Knowledge Experts | Text-to-motion retrieval | | | | Motion-to-text retrieval | | | |
|---|---|---|---|---|---|---|---|---|
| | R@1↑ | R@5↑ | R@10↑ | MedR↓ | R@1↑ | R@5↑ | R@10↑ | MedR↓ |
| No Part Text | 9.87 | 26.36 | 37.38 | 19.50 | 10.19 | 26.56 | 36.63 | 21.50 |
| LLaVA1.6-34B [17] | 11.82 | 28.36 | 41.89 | 16.00 | 12.18 | 28.62 | 39.93 | 17.00 |
| LLaMA3-70B [37] | 12.86 | 30.75 | 43.00 | 15.00 | 13.82 | 30.09 | 41.83 | 16.00 |

Table 4: Comparison of the computational complexity and runtime using one NVIDIA 4090 GPU.

| Method | ChronRet [6] | MoPatch [45] | SGAR | SGAR++ |
|---|---|---|---|---|
| **FLOPs** | 5.48G | 6.46G | 6.89G | 6.89G+1.05G*5=12.12G |
| **Runtime** | 0.0072s | 0.0078s | 0.0083s | 0.0086s |

**1) Effectiveness of Part Motion Modeling.** The crucial idea of our method is the integration of the structural part motion learning. Specifically, it consists of two parts, *i.e.*, the basic part contrastive learning for subtler semantic alignment and the part-mixture learning for better local motion decorrelation. As shown in Table 2, it significantly boosts the model performance, demonstrating stronger human motion understanding capacity due to the fine-grained action unit modeling.

**2) Effectiveness of Directional Relation Regularization.** Meanwhile, the proposed relational consistency further regularizes the feature manifold and pursues the corresponding relation knowledge consistency, which improves the model representations as shown in Table 2.

**3) Different LLMs for Generation.** Table 3 shows the results using different LLMs for part description generate. First, it can seen the above LLM choices can generally be bring about performance improvement, which indicates the promising future of our GAR paradigm. Besides, we find the general reasoning capacity of LLMs matters more than the visual capacity in our agent task. Furthermore, employing stronger LLMs could further improve part text quality and performance, which, however, is not the focus of this work. We kindly refer the readers to Appendix A for more discussions.

**4) Complexity Analysis.** As shown in Table 4, our SGAR has similar complexity and runtime to other methods. When testing GAR is enabled (SGAR++), the main additional cost is the encoding of the extra part texts (1.05G for once forward of text encoder, here we take 5 parts for example). In terms of the runtime, it should be noted that the encoding of part texts is highly parallel and does not incur much time cost.

## 5 Applications

In the following, we demonstrate the transfer capacity and application value of our model in terms of the motion encoder and text encoder.

### 5.1 Skeleton-Based Human Motion Understanding

**1) Transfer Learning for Action Recognition.** We adopt the BABEL 60-classes benchmark [27] consisting of 10892 sequences from AMASS. It is challenging due to the extremely short clips and fine-grained action categories. In our transfer learning setting, the motion encoder is pre-trained on HumanML3D/Motion-X first, and then transferred to the BABEL dataset for 60-class action recognition. A new fully-connected (FC) layer is attached to the encoder to predict final scores. As shown in Table 5, our method can surpass the state-of-the-art supervised models by fine-tuning the last FC layer solely. When we utilize the larger dataset, *i.e.*, Motion-X, a stronger transfer learning capacity can be achieved. Finally, we demonstrate the best performance by fine-tuning the whole model, indicating the promising benefits for skeleton-based action recognition.

**2) Cross-Dataset Motion Retrieval.** In this part, we implement the cross-dataset retrieval, *i.e.*, training the model on HumanML3D first and then testing on Motion-X test set, which covers various domains and noise as a more comprehensive benchmark.

Table 5: Transfer learning results for action recognition on BABEL benchmark. "L" indicates linear evaluation and "F" uses fully finetuning for evaluation.

| Method | Pretrain | Finetune | Top-1 | Top-5 |
|---|---|---|---|---|
| 2s-AGCN [34] | - | F | 41.14 | 73.18 |
| MotionCLIP [35] | - | F | 40.90 | 57.71 |
| TMR [25] | HML3D | L | 40.16 | 70.42 |
| MoPatch [45] | HML3D | L | 41.05 | 71.97 |
| SGAR | HML3D | L | 42.59 | 73.09 |
| SGAR | Motion-X | L | **43.05** | **73.82** |
| SGAR | Motion-X | F | **46.21** | **77.09** |

Table 6: Cross-dataset motion retrieval results. All models are trained on HumanML3D dataset.

| Method | R@1 | R@5 | R@10 | MedR |
|---|---|---|---|---|
| *T-to-M Retrieval* | | | | |
| TMR [25] | 20.23 | 46.18 | 60.95 | 8.38 |
| MoPatch [45] | 24.00 | 48.81 | 64.32 | 7.50 |
| SGAR | **31.05** | **54.77** | **69.22** | **6.81** |
| *M-to-T Retrieval* | | | | |
| TMR [25] | 20.21 | 43.82 | 59.05 | 9.24 |
| MoPatch [45] | 22.68 | 46.45 | 61.74 | 8.25 |
| SGAR | **30.13** | **52.92** | **66.77** | **7.46** |

Table 7: Results (Top1-accuracy) of using different text encoders for zero-shot skeleton-based action recognition.

| Method | NTU 60 split | | NTU 120 split | |
|---|---|---|---|---|
| | 55/5 | 48/12 | 110/10 | 96/24 |
| SA-DVAE [13] | | | | |
| Sentence-BERT [30] | 82.37 | 41.38 | 68.77 | 46.12 |
| CLIP-B [28] | 83.28 | 39.72 | **73.88** | 48.38 |
| CLIP-L [28] | 79.35 | 36.39 | 70.21 | 49.99 |
| Distill-BERT [32] | 83.78 | 39.72 | 55.51 | 47.94 |
| ChronRet (Distill-BERT) [6] | 31.49 | 11.47 | - | - |
| LaMP (BERT) [14] | 38.05 | 14.92 | - | - |
| SGAR (Distill-BERT) | **83.92** | **45.66** | 69.97 | **52.27** |

Table 8: Quantitative results of motion generation with different text encoders.

| Method | R-Precision ↑ | | | FID ↓ | MM-Dist ↓ |
|---|---|---|---|---|---|
| | Top-1 | Top-2 | Top-3 | | |
| MDM [36] | | | | | |
| CLIP [28] | - | - | 0.611 | 0.544 | 5.566 |
| TMR [25] | 0.459 | 0.663 | 0.770 | 0.528 | 3.259 |
| Lex [19] | 0.357 | 0.536 | 0.643 | 0.524 | 5.212 |
| SGAR | **0.515** | **0.710** | **0.806** | **0.221** | **3.024** |
| T2M-GPT [47] | | | | | |
| CLIP [28] | 0.491 | 0.680 | 0.775 | 0.116 | 3.118 |
| TMR [25] | 0.513 | 0.705 | 0.803 | 0.103 | 2.964 |
| LaMP [14] | **0.540** | **0.732** | **0.825** | 0.084 | **2.783** |
| ChronRet [6] | 0.528 | 0.717 | 0.810 | **0.074** | 2.915 |
| SGAR | 0.535 | 0.728 | 0.822 | 0.081 | 2.862 |

Specifically, we retrieve the counterparts from a batch following Small protocol. The results are shown in Table 6. As we can see, our method demonstrates a remarkable improvement over previous works, verifying the stronger generalization capacity. Meanwhile, we also present the model trained on Motion-X as in-domain reference, which is much better than the HumanML3D model. This is partly due to the limited domains in HumanML3D, resulting in limited generalization capacity. On the other hand, HumanML3D uses SMPL-format data and lacks hand motion supervising, leading to some motions in Motion-X being indistinguishable.

## 5.2 Stronger Text Encoder for Human Motion

**1) Application to Zero-Shot Action Recognition.** Existing zero-shot skeleton action recognition models are trained on the seen categories to generalize to the unseen categories. They necessitate a pre-trained text encoder to derive the category label embeddings, capturing the potential semantic relations between the seen and unseen action categories. To verify the effectiveness of our text representations, we vary the choice of text encoder from general trained to our human-motion-centric text encoders, and adopt the recent work [13] as the zero-shot algorithm for comparison.

The accuracy of the unseen categories is reported in Table 7. Our method can achieve better performance than other language models which are trained on general domains. This shows the advantage of our text encoder in human motion text representation.

**2) Application to Text-Guided Motion Generation.** For motion generation, the text encoder is responsible for obtaining textual embeddings as accurate guidance. However, previous works directly utilize the text encoder derived from vision-language pre-training, *e.g.*, CLIP. Here we study the motion generation from the perspective of text embedding quality which few previous works consider. Similarly, we utilize the text embeddings output by different text encoders, and adopt two popular generation methods as baseline, the diffusion-based MDM [36] and the auto-regressive model T2M-GPT [47]. Following the previous works [8, 36], the evaluation metrics include Frechet Inception Distance (FID), Top 1-3 retrieval precision (R-Precision), and Multi-modal Distance (MM-Dist).

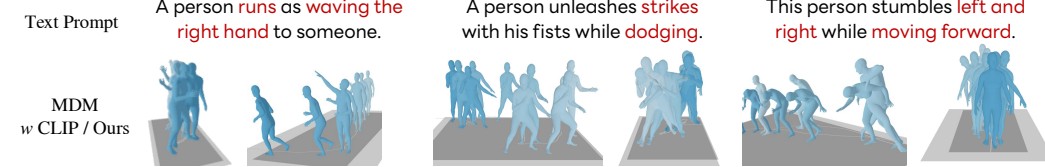

Figure 4: Comparison of motion generation results guided by different text encoders. Left: MDM model with default CLIP text encoder. Right: MDM model with our text encoder.

Specifically, we compare different text encoders for motion generation. As shown in Table 8, we can find existing motion-language pre-trained text encoders perform better than traditional CLIP baseline for motion generation tasks due to human-centric motion modeling, and our text encoder achieves competitive scores compared with other latest methods (Note that LaMP utilizes a larger text encoder, *i.e.*, BERT). Meanwhile, evidenced by the cases in Figure 4, our text encoder can output more accurate text embeddings as conditions due to the decoupled structural learning, effectively alleviating the difficulty of composite motion generation.

## 6  Conclusion

This paper presents a novel structural generative augmented retrieval paradigm for motion-text data, which exploits part motion concepts as more generalizable knowledge to boost retrieval in a GAR manner. By generating part linguistic knowledge through LLMs, SGAR effectively aligns part motions with part-mixture decorrelation learning and directional relation consistency techniques. Remarkably, the adopted GAR paradigm enables our model with an optional testing-augment capacity for fine-grained and improved retrieval. Extensive experiments on retrieval as well as recognition, and generation applications, demonstrate our superior performance and promising transferring capacity.

## 7  Acknowledgements

This work was supported in part by the National Key R&D Program of China (No. 2024YFB2809101), and in part by the National Natural Science Foundation of China under Contract No.62172020.

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

# A  Discussion on Structural Knowledge Generation by LLMs

## A.1  Comparison of Different LLMs for Generation.

Table 9: Comparison for the different knowledge experts for part motion knowledge generation.

| Knowledge Experts | Text-to-motion retrieval | | | | Motion-to-text retrieval | | | |
|---|---|---|---|---|---|---|---|---|
| | R@1 ↑ | R@5 ↑ | R@10 ↑ | MedR ↓ | R@1 ↑ | R@5 ↑ | R@10 ↑ | MedR ↓ |
| No Part Text | 9.87 | 26.36 | 37.38 | 19.50 | 10.19 | 26.56 | 36.63 | 21.50 |
| LLaVA1.6-34B [17] | 11.82 | 28.36 | 41.89 | 16.00 | 12.18 | 28.62 | 39.93 | 17.00 |
| LLaMA3-70B [37] | 12.86 | 30.75 | 43.00 | 15.00 | 13.82 | 30.09 | 41.83 | 16.00 |
| ChatGPT-3.5 Turbo [2] | 12.47 | 30.84 | 42.09 | 15.00 | 13.48 | 29.89 | 41.07 | 16.00 |
| Hybrid | 12.61 | 29.14 | 43.92 | 15.00 | 13.75 | 28.96 | 40.29 | 16.50 |

We compare the widely used MLLM/LLM model, LLaVA model (v1.6-34B) [17], LLaMA3-70B [37], and GPT-3.5 Turbo [2] model. For LLaVA, we render 3D motions into a sequence of images, which are then stitched together as additional visual prompts. In our experiments, we observe that LLaVA tends to over-rely on the image prompt, leading to confusion between the left and right sides. To mitigate this, we use distinct colors for the left and right sides to aid the model's differentiation. However, the hallucination problem still exists. Some examples are shown in Figure 5, where LLaVA still exhibits hallucinations with limited output diversity. We also present the diversity results quantitatively in Table 10 by calculating the type-token ratio of the generated part descriptions in the HumanML3D dataset to support our claim.

We provide the quantitative results of using different LLMs as shown in Table 9. In addition to relying on a single LLM, we have explored refining the LLaVA results by LLaMA to combine their merits, denoted as "Hybrid". Specifically, given the global texts and the part texts generated by LLaVA, we prompt the LLaMA to improve the accuracy and diversity of these part texts. However, we do not observe a non-trivial improvement as shown in Table 9.

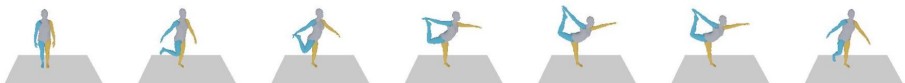

*A man grabs his right leg with right hand and extends left arm into an akward stretch.*

#LLaVA Output:
- The right arm is extended outward and upward, then bent at the elbow and brought toward the body, then extended outward again.
- The left arm is extended outward and upward, then bent at the elbow and brought toward the body, then extended outward again.
- The right leg is extended straight out to the side, then bends at the knee and brings the foot toward the body, then extends straight out again.
- The left leg is extended straight out to the side, then bends at the knee and brings the foot toward the body, then extends straight out again.
- The torso is upright and facing forward, then leans to the side as the arms are brought in, then straightens as the arms extend outward again.

# LLaMA Output:
- The right arm bends downward from straight to bent when grabbing the right leg.
- The left arm extends from bent to straight when stretching outward.
- The right leg is lifted upward by the right hand.
- The left leg remains stationary, providing balance.
- The torso leans forward slightly, aligning with the direction of the stretch.

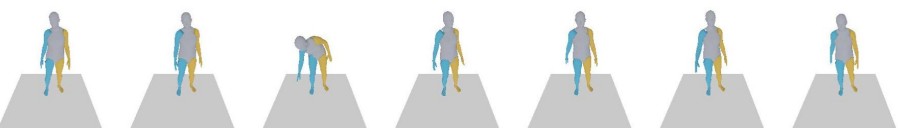

*A person picks up something with his right hand, then raises it slightly several times.*

#LLaVA Output:
- The right arm is extended downward and then retracted upward.
- The left arm is extended downward and then retracted upward.
- The right leg is extended forward and then retracted backward.
- The left leg is extended forward and then retracted backward.
- The torso is bent forward and then straightened back up.

# LLaMA Output:
- The right arm bends from straight to bent when reaching downward and then extends from bent to straight when lifting the object.
- The left arm remains relaxed under the shoulder by the side of the body.
- The right leg is stationary, providing balance.
- The left leg is stationary, providing balance.
- The torso leans forward slightly when reaching downward and then returns to upright when lifting the object.

Figure 5: Some examples of the part motion text generation results of LLaVA and LLaMA. The output descriptions that differ greatly from the actual motions are highlighted in red.

Finally, we select LLaMA3-70B for linguistic knowledge generation, as we find that its superior reasoning capacity is more critical than additional visual prompts for our task. Another reason is that in the context of GAR, paired motion-text data is often unavailable during testing. Therefore, jointly utilizing the motions and texts as prompts, such as LLaVA, is not preferred. Although this joint strategy can be considered to enhance the text quality during training, we find that a general text-only generation strategy can already achieve satisfactory performance.

**Overall, our paradigm is compatible with different generation strategies, which is not the very focus of this work. We believe that as the capabilities of future LLMs improve, the proposed GAR-based solutions are promising.**

Next, we further discuss the generation quality and model robustness to support our motivation.

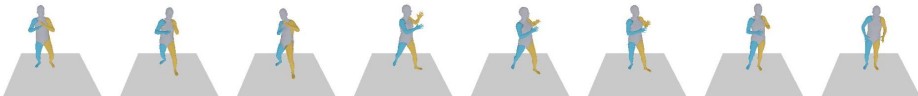

Input Motion Text: A person holds both hands in front of them, punching forward with the left hand, then punching to the left hand side before stopping.
# LLaMA Output:
- The right arm is holding straight in front, then moving to the right (when punching to the left hand side) and stopping.
- The left arm is holding straight in front, bending at the elbow when punching forward, and then extending when punching to the left.
- The right leg is [udf].
- The left leg is [udf].
- The torso is [udf].

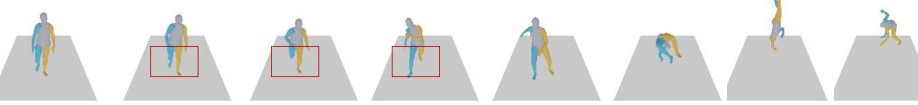

Input Motion Text: A person breakdances while kicking feet and then flips backwards while spinning on their hands, landing on their feet.
# LLaMA Output:
- The right arm is bending from straight to bent when placing hands on the ground, and then extending from bent to straight when pushing off ground to flip backwards.
- The left arm is bending from straight to bent when placing hands on the ground, and then extending from bent to straight when pushing off the ground to flip backwards.
- The right leg is kicking outward, lifting off the ground, and then tucking inward when flipping backwards, and finally landing on the ground. #missing walking forward
- The left leg is kicking outward, lifting off the ground, and then tucking inward when flipping backwards, and finally landing on the ground. #missing walking forward
- The torso is twisting rapidly when spinning on hands, and then aligning with the direction of the flip when landing on feet.

Figure 6: Some examples of the imperfect generated part descriptions. For example, the right arm "holding straight in front" is not accurate in the first case. And in the second case, the man walks forward first as shown in the rendered images while the leg motion texts miss this movement. However, these imperfect part texts, due to the lack of visual information or incomplete global motion, are still closely related to the part motions to benefit the model training.

Table 10: Type-Token Ratio analysis of the generated part motion descriptions by LLaVA and LLaMA in HumanML3D dataset. A higher metric indicates more diverse corpus.

| Knowledge Experts | Type-Token Ratio ↑ |
|---|---|
| LLaVA1.6-34B | 0.360 |
| LLaMA3-70B | **0.416** |

## A.2 Discussion on the Quality of Generated Texts and Model Robustness.

Based on our observation, most LLM-generated part descriptions effectively capture local motion patterns with clear kinematic phrasing. Although a subset of results (particularly for complex motions, as illustrated in Figure 6) exhibit minor deviations from ground truth, they are acceptable and still closely associated with the actual movements, which usually deviate only in some small movements. With the supervision of the accurate ground-truth global motion descriptions, the additional part texts introduce structurally decomposed knowledge as regularization. Therefore, the generated part texts can still serve as beneficial knowledge for model training, as evident in Table 9.

On the other hand, we study the model robustness on the quality of the generated part texts. We artificially perturb the quality of the part texts for a portion of the training data to roughly simulate the imperfections in the part texts generated by LLMs. Specifically, we randomly select and exchange the part texts of different data samples, or randomly delete a certain proportion of words, or randomly add some predefined motion descriptions of part movements as noise. In implementation, to eliminate

Table 11: Retrieval results using artificially perturbed part texts for training. Noise ratio $\alpha$ means the part texts of $\alpha$ ratio of training data are perturbed through human intervention to roughly simulate the imperfections in the part texts generated by LLMs.

| Noise Ratio | Text-to-motion retrieval | | | | Motion-to-text retrieval | | | |
|---|---|---|---|---|---|---|---|---|
| | R@1 | R@5 | R@10 | MedR | R@1 | R@5 | R@10 | MedR |
| No Part Training | 9.87 | 26.36 | 37.38 | 19.50 | 10.19 | 26.56 | 36.63 | 21.50 |
| 0% | 11.27 | 28.44 | 41.21 | 16.50 | 12.57 | 27.87 | 39.74 | 19.00 |
| 10% | 11.06 | 28.34 | 40.80 | 16.50 | 12.87 | 27.80 | 38.92 | 19.50 |
| 20% | 10.45 | 27.07 | 39.05 | 18.00 | 12.21 | 27.04 | 37.49 | 19.50 |
| 30% | 10.62 | 26.41 | 38.24 | 19.00 | 11.12 | 26.38 | 36.92 | 21.00 |

the influence of other proposed alignment strategies, we only use global and local contrastive learning objectives for training. The results are shown in Table 11. As we can see, our paradigm is robust to the part text quality. When the noise ratio is less than 30%, it can always show a non-trivial performance improvement compared to training without part involvement. This is mainly because under the dominant global supervision, slight noise from body parts will not make the model to learn "wrong" representations. Meanwhile, due to the relatively low diversity of body part motions, this adverse effect is further weakened. Instead, the knowledge of body parts can be used as an effective regularization to guide the precise alignment learning of the model, thereby achieving performance improvement.

## B    Visualization Results

### B.1    Motion-Text Retrieval Comparison with Other Methods

We present more motion-to-text and text-to-motion retrieval results in Figure 8 and Figure 7 for comparison. As we can see, owing to the part motion knowledge modeling, our method can retrieve results that are more consistent at fine-grained semantics, *e.g.*, "at knee height" (the third case in Figure 7), "in a circle" (the third case in Figure 8). This is not observed in the baseline method MoPatch [45], which solely applies the global alignment, leading to the potential over-fitting with limited data.

### B.2    Motion-Text Retrieval Results with/without Testing GAR

To better illustrate the effect of our proposed testing-augment policy, *i.e.*, GAR using additional part information generated by LLMs, we present some qualitative results of text-to-motion retrieval results with/without Testing GAR in Figure 9.

As we can see, providing additional part information is generally beneficial for retrieval during testing. More precisely, the "generation" of part information is more like a kind of information reformulation to explicitly associate the motion patterns with the linguistic verbs. Although part motion information is derived from full-body motion descriptions, exploiting it can still bring about improvement. We believe that this improvement mainly comes from a more precise local concept matching constraint. This leads to the explicit retrieval of local motion concepts. For example, the model without testing GAR can generally retrieve the roughly similar motions according to the queries in Figure 9. However, it is still insufficient for the precise matching of some local motion patterns. Especially when dealing with some long text queries, some movement details can be ignored by the text encoder. At this time, the decoupling of structural parts can effectively enhance the loyalty of retrieval results to this query information.

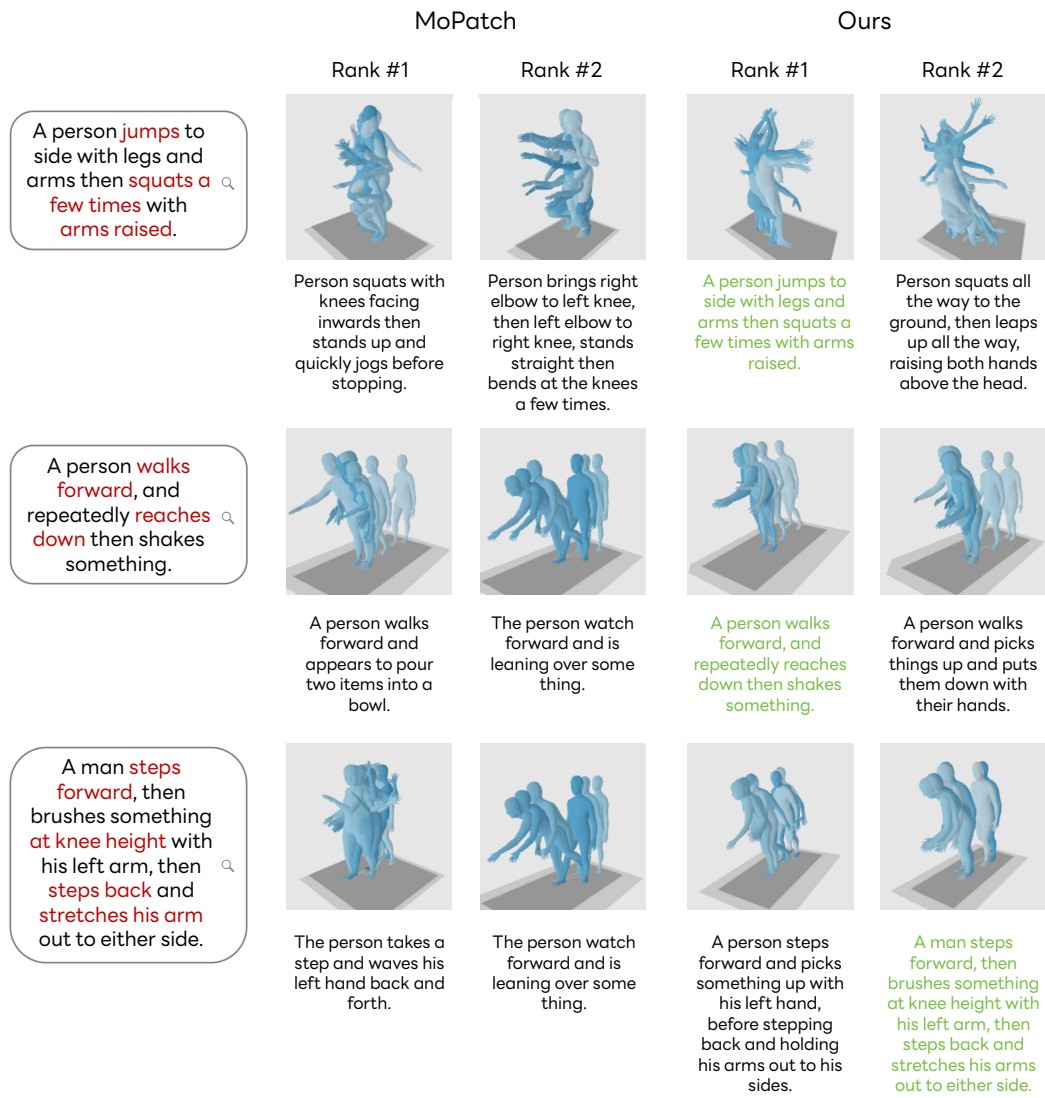

Figure 7: Comparison results of text-to-motion retrieval between MoPatch [45] and our proposed SGAR. The top-2 retrieval motion results with their textual annotations are shown here. Green texts indicate the perfect match of the query text. All motions in the gallery are from the test set and were unseen during training.

MoPatch                                    Ours

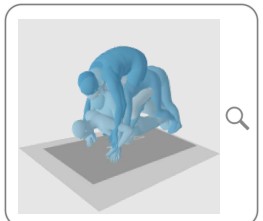

**Rank #1**: A man bends over and puts his hands on the ground and is on all fours.
**Rank #2**: The figure rises from a laying position and walks in a counterclockwise circle, and then lays back down the ground.
**Rank #3**: A figure is laying on the ground then rolls into a sit, then into being on all fours, then to standing.
**Rank #4**: The person was laying down and then they got up.

**Rank #1**: A man does a push up and then uses his arms to balance himself back to his feet.
**Rank #2**: A person doing torso-twists, side-bends, and a lower back/hamstring stretch.
**Rank #3**: A person who is prone pushes himself up off the ground using his arms and propping himself with his knees before standing awkwardly in a ready to wrestle position.
**Rank #4**: The person was laying down and then they got up.

**Ground-Truth Text**: A man does a push up and then uses his arms to balance himself back to his feet.

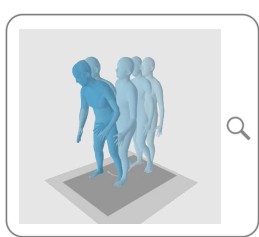

**Rank #1**: A person walks forward and appears to pour two items into a bowl.
**Rank #2**: A man steps forward, then picks something up with his right hand, then with his right hand, brings them close together, and sets them back down in the same order.
**Rank #3**: A man steps forward, then picks something up with his left hand, then with his left hand, brings them close together, and sets them back down in the same order.
**Rank #4**: A person walks forward and moves something with his right hand.

**Rank #1**: This person steps forward and grabs an item then moves his left arm up and down.
**Rank #2**: Person walks forward with left hand extended to side, trying to feel something.
**Rank #3**: This person steps forward and grabs an item then moves his right arm up and down.
**Rank #4**: A person gracefully walks forward, picks up an object, and raises it towards their face.

**Ground-Truth Text**: This person steps forward and grabs an item then moves his left arm up and down.

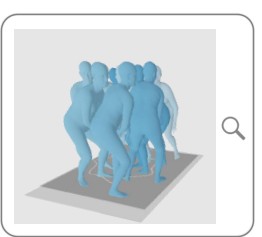

**Rank #1**: A person is crouched down and walking around sneakily.
**Rank #2**: A person with both feet on the ground with both knees bended moving from one side to another, trying to hide or something.
**Rank #3**: A man slowly leans forward and moves around and carries a heavy object.
**Rank #4**: A person is sneaking around.

**Rank #1**: A person limping with left leg hurt and going around in a circle.
**Rank #2**: A person limping with right leg hurt and going around in a circle.
**Rank #3**: A person walks in a counterclockwise circle while bent forward at the waist, and holding their left leg with both hands.
**Rank #4**: A person walks in a clockwise circle while bent forward at the waist, and holding their right leg with both hands.

**Ground-Truth Text**: A person walks in a counterclockwise circle while bent forward at the waist, and holding their left leg with both hands.

Figure 8: Comparison results of motion-to-text retrieval using our method with and without Testing GAR.

## C    Additional Experimental Results

### C.1    Retrieval Based on Only Part Queries

In the previous sections, we primarily utilize the part information as the auxiliary information of the full-body motion. Next, we study the motion retrieval performance using only the part-based queries, *i.e.*, no full-body motion descriptions.

As shown in Table 12, retrieval only relying on part queries is confusing due to the absence of other part motion descriptions. As shown in Figure 3, our model can return reasonable results that differ in

Figure 9: Text-to-Motion retrieval results in terms of using or not using testing GAR, *i.e.*, jointly considering the full-body motion-text similarity and the part motion-text similarity to derive the final retrieval results.

Table 12: Text-to-motion retrieval results by only part queries on HumanML3D benchmark.

| Query Type | Text-to-motion retrieval | | | | | |
|---|---|---|---|---|---|---|
| | R@1 | R@2 | R@3 | R@5 | R@10 | MedR |
| Left Arm | 1.92 | 3.06 | 4.63 | 7.17 | 12.12 | 171.00 |
| Right Arm | 2.46 | 3.81 | 5.18 | 8.06 | 12.87 | 150.00 |
| Torso | 3.54 | 5.27 | 7.12 | 10.20 | 16.91 | 89.00 |
| Left Leg | 2.58 | 3.63 | 5.39 | 8.31 | 13.90 | 154.50 |
| Right Leg | 2.81 | 3.90 | 5.91 | 9.01 | 14.67 | 127.50 |
| All Parts | 6.21 | 9.63 | 13.16 | 18.18 | 27.18 | 37.00 |

the global motion but match the query for local motion patterns, which can support partial match retrieval [29]. We also study the retrieval by averaging several types of part information to jointly perform retrieval as shown in Table 12. However, the performance is still lower than using full-body motion descriptions. This is because when decoupled to local part motion descriptions, it is difficult for the query to contain global motion semantics and the temporal correspondence, resulting in degraded retrieval results. Therefore, our key idea is to use part information as auxiliary information for retrieval or to enable the possibility of partial match retrieval.

## C.2 Different ViT Backbones

Table 13 shows the results of different sizes of the ViT backbone. As the size of the model increases, the performance improves. Note that this is not observed in the previous works [25, 45], where the model still suffers from severe over-fitting problem, resulting in poorer performance of ViT-L than ViT-B. In contrast, our model well exploits the more general part knowledge to enable better generalization, alleviating the over-fitting problem with limited training data.

Table 13: Ablation studies of different ViT backbone sizes on HumanML3D benchmark.

| ViT Size | Text-to-motion retrieval | | | | Motion-to-text retrieval | | | |
|---|---|---|---|---|---|---|---|---|
| | R@1 | R@5 | R@10 | MedR | R@1 | R@5 | R@10 | MedR |
| Small | 11.79 | 28.85 | 41.61 | 16.00 | 12.64 | 29.13 | 39.94 | 17.00 |
| Base | 12.86 | 30.75 | 43.00 | 15.00 | 13.82 | 30.09 | 41.83 | 16.00 |
| Large | 13.05 | 31.84 | 44.32 | 14.00 | 13.73 | 31.18 | 42.01 | 16.00 |

### C.3 More Ablation Study Results

We present the ablation results of the hyper-parameters, $\lambda_{mix}$ and $\lambda_r$ as the loss weight of the mixing objectives in Table 15, and the relational regularization in Table 14. We choose the best settings for implementation.

## D Approach and Implementation Details

### D.1 Part-Mixture De-correlation Learning

Here we provide a detailed introduction of the part-mixture learning method. First, the mixed motion sequence $m_{i,k}^{mix}$ is constructed. Specifically, giving two randomly sampled motions $m_i, m_k$, we randomly swap the patches of $2 \sim 3$ parts (5 in total) for the mixing operation. Then the mixed data are fed into the motion encoder to obtain the full-body and part motion embeddings, $z_{m,mix}$ and $\{z_{m,mix}^j\}_j$ (recall that $j$ indicates the $j_{th}$ part). Meanwhile, the text encoder takes the full-body and part textual descriptions as inputs, and outputs the corresponding embeddings $z_{t,i}$ and $\{z_{t,i}^j\}_j$. Then, we calculate the global and local alignment objectives of the mixed motions for optimization. Although the implementation is not complicated, its formalization is somewhat cumbersome, and we try to present it as clearly as possible.

For global alignment, the positive text embedding of $z_{m,mix}$ is the linear blending of $z_{t,i}$ and $z_{t,k}$, i.e., $z_{t,mix} = \lambda_m z_{t,i} + (1 - \lambda_m) z_{t,k}$, where the $\lambda_m$ is the mixing ratio determined by the number of replaced parts. Then, the global mixed alignment can be formulated as

$$\mathcal{L}_{full}^{mix,m2t} = \mathbb{E}\left[ -\log \frac{\exp(z_{m,mix} \cdot z_{t,mix}/\tau)}{\sum_j \exp(z_{m,mix} \cdot z_{t,j}/\tau)} \right],$$
$$\mathcal{L}_{full}^{mix,t2m} = \mathbb{E}\left[ -\log \frac{\exp(z_{m,mix} \cdot z_{t,mix}/\tau)}{\sum_j \exp(z_{m,j} \cdot z_{t,mix}/\tau)} \right]. \tag{5}$$

This objective introduces more diverse motion patterns for motion-language representation learning, improving the model performance.

On the other hand, we rearrange the part text embeddings according to the sampling process in mixing operation and match the part motion embeddings of the $i_{th}$ mixed motion to the corresponding part text embeddings. Denoted the rearranged part text embeddings as $\{z_{t,mix}^j\}_j$ that matches the part

Table 14: Ablation study on the loss weight of directional relation regularization.

| $\lambda_r$ | T2M R@1 | M2T R@1 |
|---|---|---|
| 0.007 | 12.73 | 13.75 |
| 0.1 | 12.86 | 13.82 |
| 0.3 | 12.61 | 13.66 |
| 0.5 | 12.25 | 13.54 |

Table 15: Ablation study on the loss weight of part-mixture de-correlation learning.

| $\lambda_{mix}$ | T2M R@1 | M2T R@1 |
|---|---|---|
| 0.3 | 12.93 | 13.60 |
| 0.5 | 12.86 | 13.82 |
| 0.7 | 12.48 | 13.63 |
| 1.0 | 11.93 | 13.37 |

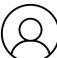

The multiple text descriptions of the same person's motion will be provided. Please decouple the specific pose and movement descriptions of different body parts from the total body motion descriptions. Default that the performer's dominant hand is the right hand. Body parts should include [right arm, left arm, right leg, left leg, torso]. If the specific movement of a body part cannot be determined, denote its motion as "undefined".

Output the description for each body part in the following format: The [body part] ... . For the output, use general words aligning with common sense and keep the sentence as simple as possible.

Q: [a man is bowling with right hand.] [a man swung his right hand backward, then threw something forward.]
A: {
The right arm is bending from straight to bent under the shoulder when preparing to release the ball and then extending from bent to straight when releasing the ball.
The left arm remains relaxed under the shoulder by the side of the body.
The right leg is undefined.
The left leg is undefined.
The torso leans forward slightly during the release of the bowling ball, aligning with the direction of the throw.
}
Q: [xxx] [xxx] ...
A:

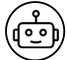

The right arm ...
The left arm ...
The right leg ...
The left leg ...
The torso ...

Figure 10: The text prompt of the part motion description generation with LLaMA3-70B.

motion embeddings $\{z_{m,mix}^{j}\}_j$, the part-mixture alignment objective is

$$\mathcal{L}_{part}^{mix,t2m} = \mathbb{E}\left[ -\frac{1}{P}\sum_{j=1}^{P} -p\left(z_{t,mix}^{j}, z_{t,*}^{j}, \tau'\right) \log p\left(z_{m,mix}^{j}, z_{t,*}^{j}, \tau\right)\right],$$

$$\mathcal{L}_{part}^{mix,m2t} = \mathbb{E}\left[ -\frac{1}{P}\sum_{j=1}^{P} -p\left(z_{m,mix}^{j}, z_{m,*}^{j}, \tau'\right) \log p\left(z_{t,mix}^{j}, z_{m,*}^{j}, \tau\right)\right], \quad (6)$$

$$where \ p_k\left(z, z_{t,*}^{j}, \tau\right) = \frac{\exp(z, z_{t,mix,k}^{j}/\tau)}{\sum\limits_{i=1} \exp(z \cdot z_{t,mix,i}^{j}/\tau)}, p_k\left(z, z_{m,*}^{j}, \tau\right) = \frac{\exp(z, z_{m,mix,k}^{j}/\tau)}{\sum\limits_{i=1} \exp(z \cdot z_{t,mix,i}^{j}/\tau)}.$$

$z_{t,mix,k}$ is the $k_{th}$ mixed sample and we have omitted $k$ in some cases where there is no ambiguity to reduce clutter.

### D.2 Experiment Settings and Implementation Details

For the motion-language alignment pre-training, we conduct the experiments on a single NVIDIA A40 GPU. We present the details of the downstream tasks as follows:

**1) Motion-Text Retrieval.** In the paper, the adopted protocols include: **a) All**: The entire test set is used as the gallery set without any modifications. This is a challenging setting because the recall scores can be directly affected by several texts with very conceptually similar meanings but different words, *e.g.*, "walk" or "walking". **b) Small Batches** (Small): This protocol is designed by Guo *et al*. [8]. It involves randomly selecting batches of 32 motion-text pairs and then reporting the average performance. While this approach introduces randomness, it serves as a benchmark for comparison. Note that the batch size of 32 is relatively manageable, making it a less challenging scenario.

For our texting-time augmentation version, SGAR++, we first calculate the similarity maps of the full-body motions and each part motions. Then, we do a weighted sum over them, and the weight ratio between the full body and any part is $10 : 1$. The final calculated similarity map is used to determine the ranks for retrieval.

**2) Transfer Learning for Action Recognition.** We process the BABEL dataset using the HumanML3D style methods to normalize the orientation, foot contact, *etc*. The pre-trained motion encoder with a new fully connected (FC) layer is fine-tuned for transfer learning. For the linear

evaluation, we only train the FC layer with the encoder fixed. The learning rate is 0.1 with a batch size of 512. For the full fine-tuning, the learning rate is 0.01 with a batch size of 384.

**3) Cross-Dataset Motion-Text Retrieval.** This is the same as Motion-Text retrieval, except for the difference of the training dataset and testing dataset. We utilize the HumanML3D dataset for training and Motion-X for testing.

**4) Zero-Shot Skeleton-based Action Recognition & Text-guided Motion Generation.** This part is mainly to evaluate the text encoder. Therefore, we simply replace the original text encoder in SA-DVAE [13], MDM [36], and T2M-GPT [47] as baseline algorithms with other text encoders to conduct the experiments. Other implementations are not changed in the baseline algorithms.

**5) Prompts for LLM Generation.** A few-shot prompting strategy is employed, with the detailed prompt shown in Figure 10.

# E    Limitations and Future Work

As discussed before, our proposed paradigm relies on the generated part texts by LLMs, which may exhibit minor imperfections for highly complex motions. As extensively analyzed in Sec. 3.2, Appendix A and Table 11, our experiments demonstrate that despite these challenges, existing LLMs can almost always be expected to boost performance, indicating the promising potential of the GAR paradigm for motion-text retrieval. Meanwhile, future work could explore temporal information generation, which remains challenging due to the lack of fine-grained temporal correspondence (*a.k.a*, weakly-supervised temporal grounding problem). This further highlights the scalability of the GAR paradigm to inspire future research.

# F    Broader Impact

Accurate human motion-text retrieval can effectively improve technologies such as virtual assistants and rehabilitation robots, and enhance the human-interaction experience. Meanwhile, it is also expected to be applied to the action demonstration in physical education teaching, dance teaching, *etc.*. Furthermore, the storage of traditional dance, martial arts, and other movements in the form of texts may help digitally preserve cultural heritage and have a positive impact. The other side of the coin is the issue of privacy protection, which needs attention to avoid being used for monitoring specific groups of people.

