# OpenReview forum: "SGAR: Structural Generative Augmentation for 3D Human Motion Retrieval"
_NeurIPS.cc/2025/Conference — NeurIPS 2025 poster_

### Official Review · Reviewer_MCsr · 2025-06-26

**Clarity:** 2
**Significance:** 3
**Originality:** 3
**Rating:** 4
**Confidence:** 4

**Summary:**

The paper handles the task of 3D human motion-to-text and text-to-motion retrieval. The main idea of the proposed solution is to leverage LLM to decompose text into detailed descriptions for different body parts. The proposed solution then aligns the text and motion representations at both global and local body part levels, which is achieved by considering a part-mixture decorrelation learning and directional relation alignment strategy.

The proposed solution outperforms existing methods regarding text-to-motion and motion-to-text retrieval.  Furthermore, the authors extend their discussion to such as action recognition, cross-dataset motion retrieval, and text-guided motion generation.

**Questions:**

1. In Tab. 1, the authors only compare the proposed solution with Lex [19] on All-HML3D protocol. What about comparisons with Lex on other protocols? Can the proposed solution consistently outperform Lex [19] across different protocols?
2. Tab. 7 discusses text-guided motion generation. Still, how does the proposed method perform against state-of-the-art text-guided motion generation methods? Furthermore, I noticed that LaMP and ChronNet also study text-motion retrieval and discuss motion generation. What are the results when comparing the proposed method with LaMP [14] and ChronNet [7] on motion generation?
3. The t-SNE visualization in Fig. 10 is confusing. Why is the proposed method considered better in terms of visualization? Is it because the line segments are generally shorter? Still, for the figure of "Our-right arm part," the paired data points present significant distances.

**Ethical Concerns:**

["NO or VERY MINOR ethics concerns only"]

**Final Justification:**

I appreciate the feedback from the authors. I consider that this feedback could address most of my concerns raised in my original comments. As a result, I will keep the original score.

**Limitations:**

Yes the authors have addressed the limitations and potential negative social impact.

**Paper Formatting Concerns:**

No major formatting issues in this paper.

**Quality:**

3

**Strengths And Weaknesses:**

**Strength**
- The paper provides comprehensive experiments, demonstrating competitive performance in text-to-motion and motion-to-text retrieval. Extension to diverse application tasks (Sec. 5) strengthens the submission.
- The proposed part-mixture decorrelation learning and directional relation alignment strategy effectively aligns the text and motion representation at both global and part levels.

**Weakness**
- The idea of leveraging LLM to decompose text descriptions into body parts has been previously discussed in motion-text literature [1], where the submission fails to discuss this literature.
- I still have some concerns related to the comparisons, as listed in the question session. I hope the authors could help address these questions.

[1] CoMo: Controllable Motion Generation through Language Guided Pose Code Editing. ECCV'24


I vote for a "borderline accept" for the current version. Although decomposing motion descriptions with LLMs is not novel, the proposed part-mixture decorrelation learning and directional relation alignment strategy is interesting.
Moreover, in the main tasks of motion-text retrieval, the proposed solution generally demonstrates competitive performance, with ablation studies verifying the key designs.

---

> ### Author Rebuttal · Authors · 2025-07-31
>
> We thank the reviewer for the valuable and positive feedback with constructive suggestions! We hope our responses adequately address the following questions raised about our work. Please let us know if there is anything we can clarify further.
>
> ---
>
> **W1: Discussion on the previous work [1]**
>
> Thanks for the helpful suggestions. While [1] also uses LLM to decouple body motion into part motions, we emphasize the following differences:
>
> - The motivations and objectives are different. The work [1] focuses on the motion generation task, aiming to enhance model generation performance through part motion conditioned generation, which can serve for fine-grained controllable generation. In contrast, we target motion-text retrieval and cross-modal alignment learning, aiming to learn a shared semantic space through explicit part decoupling to mitigate potential over-fitting problems.
> - In terms of the LLM knowledge generation, [1] adopts a complex prompting strategy, i.e., PoseScript+LLM, which we have thoroughly considered and discussed (Line 118-123). To our knowledge, PoseScript, as a non-deep learning method, is highly sensitive to hyperparameters and exhibits fragile performance across different motion domain data, significantly limiting its practical application potential. On the other hand, the PoseScript+LLM generation strategy requires the simultaneous availability of motion and text data, which is only possible during the training stage and contradicts our proposed GAR paradigm. In comparison, we employ a more universal and generalized strategy of few-shot + special token generation only dependent on texts, enabling our model to perform GAR augmentation during the testing stage.
> - The utilization of part text data differs. [1] employs the traditional generative condition injection learning approach, i.e., auto-regressive generation, whereas we propose part mixture decorrelation learning and directional alignment to fully leverage part information for alignment.
>
> We will add this discussion in the final version.
>
> [1] CoMo: Controllable Motion Generation through Language Guided Pose Code Editing, ECCV 2024.
>
> ---
>
> **Q1: More comparison with Lex**
>
> Thanks for your advice. Lex [19] is the recent work, while the authors do not make the code open-sourced. Due to its complex training stages and designs, it is difficult to reproduce it well. Therefore, we have to directly copy the reported results of Lex for comparison (**only All protocol is reported in the Lex paper**). We have provided the additional comparison of All protocol on the KIT-ML dataset in the following table, from which we can see he effectiveness of our method, especially for the motion-to-text retrieval. *Meanwhile, from an empirical perspective, the All protocol is well representative of the results of other protocols.*
>
> Table 1: Comparion with Lex on motion-text retrieval.
> | Dataset   | Method | (T2M) R@1 | (T2M) R@5 | (T2M) R@10 | (T2M) MedR | (M2T) R@1 | (M2T) R@5 | (M2T) R@10 | (M2T) MedR |
> |-----------|--------|-----------|-----------|------------|------------|-----------|-----------|------------|------------|
> | HumanML3D | Lex    | 11.80     | 30.81     | 43.36      | 14.00      | 12.39     | 29.25     | 40.34      | 17.00      |
> |           | **SGAR** | **13.51** | **31.75** | **44.75**  | **14.00**  | **14.33** | **30.92** | **42.23**  | **15.00**  |
> | KIT-ML    | Lex    | 15.13     | 36.81     | 54.12      | 8.00       | **15.01**     | 35.63     | 47.53      | 10.50      |
> |           | **SGAR** | **15.42** | **41.33** | **55.90**  | **8.00**   | 14.70 | **37.11** | **50.60**  | **9.00**   |
>
> ---
>
> **Q2: More comparison on motion generation with LaMP and ChronNet**
>
> Thanks for your question. We have provided additional results on motion generation, including LaMP and ChronNet with one of the recent SOTA methods, MoMask [1], as baseline. As shown in the following tables, our method can benefit the current motion generation methods significantly. Moreover, compared to ChronNet, our method has overall better performance. While compred to LaMP (*Note LaMP adopts the larger text encoder, which can lead to unfair comparison*), our method can achieve better FID scores with smaller DistillBERT text encoder.
>
> Table 2: Comparison with MoMask and T2MGPT as the backbones.
> | Method          | R@1   | R@2   | R@3   | FID   | MM-Dist |
> |-----------------|-------|-------|-------|-------|---------|
> | MoMask          |      |      |      |      |        |
> | Baseline (CLIP) | 0.524 | 0.715 | 0.809 | 0.073 | 2.965   |
> | LaMP (BERT)     | 0.546 | **0.750** | **0.843** | 0.058 | **2.792**   |
> | ChronNet (DistillBERT)       | 0.541 | 0.737 | 0.838 | 0.056 | 2.885   |
> | SGAR (DistillBERT)       | **0.550** | 0.748 | 0.842 | **0.049** | 2.864 |
> | T2M-GPT         |      |      |      |      |        |
> | Baseline (CLIP) | 0.491 | 0.680 | 0.775 | 0.116 | 3.118   |
> | LaMP (BERT)  | **0.540** | **0.732** | **0.825** | 0.084 | **2.783**   |
> | ChronNet (DistillBERT)     | 0.528 | 0.717 | 0.810 | **0.074** | 2.915   |
> | SGAR (DistillBERT)   | 0.535 | 0.728 | 0.822 | 0.081 | 2.862 |
>
> **1) For MoMask, we made efforts to reproduce MoMask during rebuttal with the official code. However, we cannot reproduce the performance reported in the original paper in terms of FID metric (0.073 vs. 0.045). We have found that a few researches also encountered similar problems posted in the github issue. Therefore, during rebuttal, we report the currently reproduced results for reference.*
> *2) LaMP utilizes a larger text encoder, i.e., BERT, while ChronNet and our method SGAR use the smaller DistillBERT.*
>
> We also provide more comprehensive comparison of LaMP and ChronNet on motion understanding and zero-shot action recognition in the **W4 of Reviewer pgZA** to demonstrate the advantages of our method.
>
>
> [1]MoMask: Generative Masked Modeling of 3D Human Motions, CVPR 2024.
>
> ---
>
> **Q3: Explain t-SNE visualization**
>
> Sorry for the confusion. We make efforts for cross-modal feature visualization to provide more intuitive comparison results, although this might not be easy. As explainations, for the left two figures, it is to demonstrate our method has a more aligned motion-text representation space because the line segments (connecting paired motion-text data) are shorter. Then, for the third column figure, it can be viewed as an ablation study to demonstrate the difficulty in performing retrieval only based on part information caused by the less diverse texts and false negative problem in contrastive learning. Therefore, we can conclude that only part-level alignment without global information is insufficient to achieve desirable retrieval performance, as verified in Table 11. Based on this, our idea is to use the part information as a beneficial supplement to enhance the global alignment during testing.
>
> Sorry for the confusion again. We are considering some better presentations, e.g., replacing this figure with the feature space of full body+right-arm in the final version to better demonstrate the improvement effect of part-level alignment on the feature space. We will refine this in our final version.

---

> > ### Comment · Reviewer_MCsr · 2025-08-05
> >
> > I thank the authors for providing the rebuttal. I consider that most of my concerns have been addressed. I am glad to keep a positive score.

---

### Official Review · Reviewer_pgZA · 2025-06-29

**Clarity:** 2
**Significance:** 3
**Originality:** 3
**Rating:** 4
**Confidence:** 4

**Summary:**

This paper presents SGAR, a novel framework for 3D human motion-text retrieval that leverages Structural Generative Augmentation. Rather than treating a motion as a single holistic entity, SGAR decomposes it into part-based components and uses LLMs to generate textual descriptions for each part, enabling further alignment of text and motion representations.

**Questions:**

As shown in the Weaknesses section, A reasonable explanation will make me raise the score for this work.

**Ethical Concerns:**

["NO or VERY MINOR ethics concerns only"]

**Final Justification:**

The author's response has effectively addressed my concerns. Taking into account the feedback from other reviewers, I would like to give a score of 4.

**Limitations:**

Yes

**Quality:**

3

**Strengths And Weaknesses:**

# Strengths:
- Part-based Motion Decomposition: The method breaks down human motion into smaller, more precise components (like arm movements, leg steps, etc.), providing a more granular way of aligning motion and text, while the text can be further generated by the LLMs with their prior knowledge.
- Part-Mixture Learning: A novel strategy to prevent correlation between different body parts’ motions during training by mixing parts from different motions. This encourages the model to learn more diverse and robust representations.
- Directional Relation Alignment: This approach introduces a consistency regularization that aligns full-body and part-level motion representations by modeling the directional relations between them, improving the consistency and accuracy of alignment.
# Weaknesses:
- Line 133, how much the texts with this indicator are excluded? How do the authors require the LLMs to output this indicator for the parts with high uncertainty?
- Line 166, I didn't see the Euclidean distance in formulas 1 and 2. So  how does the author ensure vector consistency while the inner product of vectors cannot guarantee that?
- Line 203, If the text encoder of SGAR include LLMs ? (IF NO: How can the human pose be split into parts during the testing?)  (IF YES: Such a huge text encoder may lead to unfairness and application.)
- In Section 5 (Application), SOTA methods such as ChronRet are not included in the comparison, even though they report relevant results in their respective papers. A more comprehensive comparison is therefore needed.
- What is k in Formula 2?

---

> ### Author Rebuttal · Authors · 2025-07-31
>
> We thank the reviewer for the valuable feedback and constructive suggestions! We hope our responses adequately address the following questions raised about our work. Please let us know if there is anything we can clarify further.
>
> ---
>
> **W1:About [udf] indicator and implementation**
>
> Thanks for your helpful question. We have calculated that ~15.9% of samples include this indicator and the corresponding texts are excluded in HumanML 3D, while ~18.2% in KIT-ML.
>
> In implementation, we utilize the designed prompt to guide the LLM to output the [udf] indicator for the parts with high uncertainty. This is inspired by previous research findings [1-5] that LLMs can effectively decompose knowledge and model its uncertainties, thereby perceiving potentially more fine-grained concepts within it. Specifically, we utilize the few-shot prompting strategy to better guide the LLM to perform this uncertainty modeling. The related prompts are (full prompts can be referred to Figure 11 in Appendix):
>
> *...
> If the specific movement of a body part cannot be determined, denote its motion as "undefined".
> Q: [a man is bowling with right hand.] [a man swung his right hand backward, then threw something forward.]
> A: {
> The right arm is bending from straight to bent under the shoulder when preparing to release the ball and then extending from bent to straight when releasing the ball.
> The left arm remains relaxed under the shoulder by the side of the body.
> The right leg is **undefined**.
> The left leg is **undefined**.
> The torso leans forward slightly during the release of the bowling ball, aligning with the direction of the throw.
> }
> Q: [xxx] [xxx] ...
> A:*
>
> [1]Large language models are versatile decomposers: Decomposing evidence and questions for table-based reasoning, SIGIR 2023
> [2]Least-to-Most Prompting Enables Complex Reasoning in Large Language Models, ICLR 2023.
> [3]Mldt: Multi-level decomposition for complex long-horizon robotic task planning with open-source large language model, arXiv 2024.
> [4]Reasoning Abilities of Large Language Models: In-Depth Analysis on the Abstraction and Reasoning Corpus, ACM TIST 2025.
> [5]A Survey of Uncertainty Estimation in LLMs: Theory Meets Practice. arXiv 2024
>
> ---
>
> **W2: Explaination on Euclidean distance in formulas 1 and 2**
>
> Sorry for the confusion. In fact, we calculate the inner product of the feature vectors in the **normalized space**. In this case, maximizing this inner product is equivalent to minimizing the l2 norm in the Euclidean space. Specifically, given two vectors x and y that are l2-normalized ($||x|| = ||y|| = 1$)
> $$ \text{argmin} \|\|\mathbf{x} - \mathbf{y}\|\|_2^2 = \text{argmin} (\mathbf{x}^\top \mathbf{x} + \mathbf{y}^\top \mathbf{y} - 2\mathbf{x}^\top \mathbf{y}) = \text{argmin} (1 + 1 - 2\mathbf{x}^\top \mathbf{y}) = \text{argmax} \ \mathbf{x}^\top \mathbf{y}.$$
> Therefore, the formulas 1 and 2 can encourage the consistency learning of positive motion-text pairs. We will clarify this in paper.
>
> ---
>
> **W3: If the text encoder of SGAR include LLMs?**
>
> Our text encoder **does not** involve an LLM. Instead, we keep the text encoder implementation the same as previous works [1-3], i.e., a pre-trained Distill-BERT, for fairness. LLMs are only for knowledge generation/augmentation as an additional external expert, which is an optional choice and independent of the text encoder (Distill-BERT). Therefore, when the LLMs are available or accessible during testing, they can be directly used for augmentation. If not, part texts can be provided by users (text-to-motion retrieval scenario), or just turn off this part-based augmentation option.
>
> [1]Exploring vision transformers for 3D human motion language models with motion patches, CVPR 2024.
> [2]TMR: Text-to-motion retrieval using contrastive 3D human motion synthesis, ICCV 2023.
> [3]Chronologically accurate retrieval for temporal grounding of motion-language models, ECCV 2024.
>
> ---
>
> **W4: More comparison results in application experiments**
>
> Thanks for your advice. We try our best during rebuttal to conduct additional experiments to include recent methods for comparison, ChronRet (ECCV 24) and LaMP (ICLR 25), following the experimental settings of Section 5. Note that ChronRet has reported the motion generation results with its text encoder and we directly report their original results. Other results are reproduced using the officially released model weights under our experimental setting.
>
> Specifically, as shown in the following tables,
> 1) ChronRet is mainly designed for temporally aware retrieval task, which does not demonstrate advantages for single action recognition. LaMP introduces a new motion-text matching task based on naive contrastive learning paradigm. However, this simple design can still not alleviate the over-fitting problem effectively, leading to sub-optimal alignment learning and generation capacity. Therefore, our method can achieve better perormance on generalized motion understanding, e.g., transfer learning or cross-dataset retrieval.
> 2) Meanwhile, we find that Lamp and ChronRet present poor performance in zero-shot action recognition with their text encoders. This indicates that these methods struggle to learn generalized text representations with limited data. In contrast, our method involves a more fine-grained part motion text alignment and regularization designs, to achieve more meaningful and generalized text representations.
> 3) For text-guided motion generation on HumanML3D dataset, we find these methods can well boost the performance compared with CLIP baseline, and our method has more advantages on R-Precision metrics compared with ChronNet (*Note LaMP adopts and finetunes a larger text encoder, BERT, which can lead to unfair comparison under a in-domain setting*).
> 4) For the main comparsion on motion-text retrieval, our method can significantly surpass ChronNet and LaMP as shown in Table 1 in the main paper.
>
> Table 1: Transfer learning results for action recognition on BABEL.
> | Method       | Finetune    | BABEL Top-1 |
> |--------------|-------------|-------------|
> | ChronRet     | Linear Head | 40.58       |
> | LaMP         | Linear Head | 38.93       |
> | **SGAR** | Linear Head | **42.59**    |
>
> Table 2: Cross-dataset motion retrieval results.
> | Method   | (T2M) R@1 | (T2M) R@5 | (T2M) R@10 | (T2M) MedR | (M2T) R@1 | (M2T) R@5 | (M2T) R@10 | (M2T) MedR |
> |----------|-----------|-----------|------------|------------|-----------|-----------|------------|------------|
> | ChronRet | 22.58     | 47.80     | 64.45      | 7.67       | 21.94     | 47.25     | 62.62      | 8.10       |
> | LaMP     | 19.48     | 46.82     | 63.28      | 7.99       | 19.46     | 42.61     | 59.49      | 9.31       |
> | **SGAR** | **31.05** | **54.77** | **69.22**  | **6.81**   | **30.13** | **52.92** | **66.77**  | **7.46**   |
>
> Table 3: Text encoder for zero-shot action recognition.
> | Method       | NTU 60 55/5 split | NTU 60 48/12 split |
> |--------------|-------------|--------------|
> | ChronRet     | 31.49       | 11.47        |
> | LaMP         | 38.05       | 14.92        |
> | **SGAR** | **83.92**   | **45.66**    |
>
> **LaMP utilizes a larger text encoder, i.e., BERT, while ChronNet and our method SGAR use the smaller DistillBERT.*
>
> Table 4: Text encoder for motion generation on HumanML3D.
> | Method       | R@1   | R@2   | R@3   | FID   | MM-Dist |
> |--------------|-------|-------|-------|-------|---------|
> | T2M-GPT      |      |      |      |      |       |
> | Baseline (CLIP) | 0.491 | 0.680 | 0.775 | 0.116 | 3.118   |
> | LaMP (BERT)  | **0.540** | **0.732** | **0.825** | 0.084 | **2.783**   |
> | ChronNet (DistillBERT)    | 0.528 | 0.717 | 0.810 | **0.074** | 2.915   |
> | SGAR (DistillBERT)    | 0.535 | 0.728 | 0.822 | 0.081 | 2.862 |
>
> **LaMP utilizes a larger text encoder, i.e., BERT, while ChronNet and our method SGAR use the smaller DistillBERT.*
>
> ---
>
> **W5: Exaplain k in Formula 2**
>
> Sorry for the confusion. In Formula 2, $p^\tau$ is a similarity distribution that consists of multiple similarity scores between $z$ and each embedding anchor, i.e., $p^\tau = \{p^\tau_1, p^\tau_2 ..., p^\tau_N\}$ where N is the batch size in our implementation. Therefore, $k$ is a subscript in Eq. (2) and $ p^\tau_k (z,z^j_{t,*}) $ denotes the similarity score between $z$  and $z^{j}_{t,k}$ after softmax with temperature $\tau$. We will polish this presentation in the final version.

---

### Official Review · Reviewer_JXyx · 2025-07-01

**Clarity:** 3
**Significance:** 4
**Originality:** 3
**Rating:** 4
**Confidence:** 3

**Summary:**

This paper proposes SGAR, a framework to improve motion-text retrieval performance by LLM-based augmentation. Specifically, it utilizes LLMs to decompose human motion descriptions into body-parts motion descriptions. It further proposes a part-mixture learning to make full use of the partitioned data, and augment training data pairs. Extensive experiment demonstrate the proposed method achieves better performance compared to previous SOTA methods. It also provides promising results in improving related downstream tasks by using the trained text/motion encoder.

**Questions:**

1. Can you explain how only using Part CL could boost the performance, which seems to be not directly related to the global semantics?

**Ethical Concerns:**

["NO or VERY MINOR ethics concerns only"]

**Final Justification:**

This work is clearly presented and meets the general quality standards of NeurIPS. However, due to the simplicity and narrow scope of the motion retrieval task, I have reservations about the broader impact and originality of the proposed LLM-based augmentation method. While there are no major flaws, the contribution is relatively modest and borders on incremental—especially in light of the declining bar for acceptance.

**Limitations:**

yes

**Paper Formatting Concerns:**

There’s no formatting concern.

**Quality:**

3

**Strengths And Weaknesses:**

## Strengths
1. The LLM-based motion decomposition is very reasonable. The proposed part-mixture learning for data augmentation is interesting and shows good improvement.
2. The Directional relation alignment strategy borrow from language modeling is also interesting and novel for the motion task.
3. The proposed augmentation naturally come up with test-time augmentation strategy that is able to boot the performance effectively as demonstrated by the quantitative experiments.

## Weaknesses
1. One major issue is the concern around reproducibility. The good performance could be better demonstrated by any kinds of video demo, interactive web demo or code that reviewers can see or test. Otherwise, it could be a bit skeptical. Another alternative is, for the major experiments, train the models with different seeds and report the average performance and error bar, since retrieval results could be sensitive to random seed at both training and testing.
2. I’m not fully convinced by the part-mixture learning design.  When mixing data sample by exchanging body parts, what is the reason to require the latent to be the interpolation of their corresponding latent with the corresponding ratio. For example, why replacing the right arm or left leg corresponds to the same semantic latent, according to the formula, if i understand correctly? Are there any related work to support this design decision?

---

> ### Author Rebuttal · Authors · 2025-07-31
>
> We thank the reviewer for the valuable feedback and constructive suggestions! We hope our responses adequately address the following questions raised about our work. Please let us know if there is anything we can clarify further.
>
> ---
>
> **W1: About reproducibility, i.e., code, performance error bars**
>
> Thanks for your helpful comments. First, the code would be available upon publication. Due to the author's policy of NeurIPS, we are sorry that we can not provide any link to a demo or code during rebuttal. Alternatively, we report the average performance with error bar to better demonstrate the performance fluctuations in the experiment under 5 runs. As mentioned, retrieval results could be sensitive to randomness, while we can see that our method can consistently achieve the SOTA performance, verifying the effectiveness of our method.
>
> Table 1: Average performance with the error bar.
> | Method | (T2M) R@1           | R@3           | R@5            | R@10          | MedR         | (M2T) R@1           | R@3           | R@5            | R@10          | MedR         |
> |--------|---------------|---------------|----------------|---------------|--------------|---------------|---------------|----------------|---------------|--------------|
> |HumanML3D |        |               |               |                |               |              |               |               |                |              |
> | MoPatch | 10.80         | 20.00         | 26.72          | 38.02         | 19.00        | 11.25         | 19.98         | 26.86          | 37.40         | 20.50        |
> | ChronRet | 8.30          | 16.70         | 23.65          | 35.31         | 22.00        | 9.08          | 11.22         | 23.68          | 34.31         | 23.00        |
> | **SGAR** | **12.82 ±0.448** | **23.08 ±0.724** | **30.72 ±0.231** | **43.45 ±0.656** | **14.20 ±0.447** | **13.71 ±0.334** | **23.41 ±0.399** | **30.45 ±0.509** | **41.57 ±0.509** | **16.00 ±0.000** |
> | **SGAR++** | **14.12 ±0.301** | **25.12 ±0.337** | **32.76 ±0.415** | **45.48 ±0.627** | **13.00 ±0.000** | **14.55 ±0.399** | **25.33 ±0.383** | **32.67 ±0.206** | **44.72 ±0.214** | **14.00 ±0.000** |
> | KIT-ML    |        |               |               |                |               |              |               |               |                |              |
> | MoPatch | 14.02         | 28.91         | 34.10          | 50.00         | 10.50        | 13.61         | 27.54         | 33.33          | 44.77         | 13.00        |
> | **SGAR** | **16.02 ±0.629** | **30.08 ±0.693** | **40.15 ±1.236** | **54.30 ±1.848** | **9.00 ±0.707** | **14.82 ±0.360** | **26.22 ±0.805** | **35.63 ±1.317** | **49.40 ±1.987** | **10.20 ±1.095** |
> | **SGAR++** | **16.27 ±0.641** | **31.49 ±0.546** | **41.04 ±1.754** | **56.83 ±2.294** | **8.00 ±1.000** | **16.71 ±0.686** | **29.72 ±0.386** | **38.83 ±0.739** | **52.73 ±1.094** | **9.00 ±0.000** |
>
> ---
>
> **W2: Further explanation on part-mixture learning design**
>
> Thanks for your valuable question. We design the part mixture learning inspired from both the success of previous image-based methods but also the characteristics of skeleton data.
>
> Specifically,
> - On the one hand, this can be referred in the image-based mixing contrastive learning works [1-3]. Similarly, they copy-and-replace the image patches like Cut-Mix, using linear interpolation of the embeddings of the images before mixing as positives for contrastive learning. Similarly, mixing ratio is adopted as the interpolation coefficient. In this case, the mentioned problems can still occur, that is, replacing image patches of the same size in the upper left corner or lower right corner may result in the same interpolated latents for contrastive learning. This seems to be an inherent defect of random mixing operations, whether it is for supervised learning or self-supervised contrastive learning. However, despite this, the success of these works has proved the effectiveness of the design. This can mainly be interpreted as an effective regularization of the feature space *in statistical sense* by applying linear equivariant transformation constraints during training [3]. To help understand, interpolation in the InfoNCE-objective-applied feature space can be viewed similar to interpolation in the prediction label space of supervised learning, (like original Cut-Mix for classification), which both perform interpolation / smoothing in the deep feature space of models.
> - In addition, previous work [4] has revealed the essence of local motion pattern modeling for skeleton data, which means a subtler meaningful semantic unit as body parts. This makes different body parts all potentially contribute to the extraction of a certain global motion concept. Therefore, the mixture at the input data level can be expected to achieve interpolated features in the high-level semantic latent space. Meanwhile, as you understand, although we do not restrict which parts to be mixed (the importance of different parts to certain global motion may vary), this design is still statistically reasonable and empirically effective as discussed above. Therefore, we adopt this simple part-wise replacement strategy, retaining the meaningful motion semantic unit while encouraging a smoother representation space as regularization. Furthermore, from the authors' personal experience, mixing augmentation improves performance mainly through the representation smoothing constraint as regularization rather than obtaining strictly precise linear interpolation coefficients.
>
> [1]i-Mix: A Domain-Agnostic Strategy for Contrastive Representation Learning, ICLR 2021.
> [2]MixSiam: A Mixture-based Approach to Self-supervised Representation Learning. arXiv 2021.
> [3]Un-Mix: Rethinking Image Mixtures for Unsupervised Visual Representation Learning, AAAI 2022.
> [4]Actionlet-Dependent Contrastive Learning for Unsupervised Skeleton-Based Action Recognition, CVPR 2023.
>
> ---
>
> **Q1: Explain why only Part CL is better**
>
> Only part CL (no LLM is used under testing) still boosts global-motion-based retrieval, which can be explained as follows:
> - The key motivation is to decompose and model the part motion consistency to benefit the general motion-text alignment learning. Taking "walks and waves hand" as an example, with single global alignment learning, the model may incorrectly correspond the motion pattern of arm swinging to the verb "walk" rather than "wave hand", especially when training with a limited data scale. This means the difficulty of motion-text retrieval arises not only from the cross-modal alignment but also from the reorganization and decomposition of semantic concepts within the same modality. In this context, learning the correspondence of local concepts as primitives can lead to more precise, generalized local alignment, and ultimately enhance global understanding.
> - It is also beneficial to introduce more diverse part texts generated by LLMs, which alleviates the over-fitting problem with more data.

---

> > ### Comment · Reviewer_JXyx · 2025-07-31
> >
> > Thanks for the response. Most of my concerns have been addressed. I will keep the positive rating.

---

> > > ### Author Response · Authors · 2025-08-01
> > > **Official Comment by Authors**
> > >
> > > Thanks for your valuable suggestions and positive support again!

---

### Official Review · Reviewer_5qdg · 2025-07-01

**Clarity:** 3
**Significance:** 3
**Originality:** 3
**Rating:** 4
**Confidence:** 4

**Summary:**

This paper presents SGAR, a novel framework for 3D human motion-text retrieval that leverages structural generative augmentation from large language models. The key innovation is decomposing holistic human motions into fine-grained semantic units (body parts) to enable more precise cross-modal alignment. The method demonstrates strong performance on motion-text retrieval benchmarks, showing significant improvements over previous state-of-the-art methods.

**Questions:**

Q1: How can we ensure that the query generated by LLM is correct or meets the search intent?
Q2: How do we explain the fact that the test is better without LLM? Is it essentially because more data (generated by LLM) is used?
Q3: This method does not seem to be suitable for action-to-text retrieval? For text-to-action, LLM can enhance text queries, but for action-to-text retrieval, LLM needs to enhance every sentence in the database, and the real-time retrieval is very poor.

**Ethical Concerns:**

["NO or VERY MINOR ethics concerns only"]

**Limitations:**

Potential error propagation from LLM hallucinations unquantified
There seems to be a limitation in the real-time nature of retrieval

**Quality:**

3

**Strengths And Weaknesses:**

S1: Well-designed ablation studies validate each component's contribution
S2: Its novel part-mixture learning and directional relation alignment effectively enhance motion representation.

W1: The discussion on computational efficiency is limited. Providing details on runtime performance and resource requirements would help assess the practical trade-offs involved.
W2: The novelty of the approach could be better contextualized by comparing more explicitly with prior part-based motion retrieval methods.
W3: How can we ensure that the query generated by LLM is correct or meets the search intent?

---

> ### Author Rebuttal · Authors · 2025-07-31
>
> We thank the reviewer for the valuable feedback and constructive suggestions! We hope our responses adequately address the following questions raised about our work. Please let us know if there is anything we can clarify further.
>
> ---
>
> **W1: Computional Efficiency**
>
> As suggested, we study the computational efficiency with FLOPs and Runtime. All experiments are conducted on one NVIDIA 4090 GPU and Intel(R) Xeon(R) Platinum 8358P CPU @ 2.60GHz.
>
> As shown in the following table, our SGAR has similar complexity and runtime to other methods. When testing GAR is enabled (SGAR++), the main additional cost is the encoding of the extra part texts (1.05G for once forward of text encoder, here we take 5 parts for example). In terms of the runtime, it should be noted that the encoding of part texts is highly parallel and does not incur much time cost.
>
> Table 1: Comparison of computational complexity and runtime.
> | Method       | FLOPs                           | Runtime  |
> |--------------|---------------------------------|----------|
> | ChronRet     | 5.48G                           | 0.0072s  |
> | MoPatch      | 6.46G                           | 0.0078s  |
> | SGAR (Ours)  | 6.89G                           | 0.0083s  |
> | SGAR++ (Ours)| 6.89G+1.05G*5=12.12G            | 0.0086s  |
>
> ---
>
> **W2: Comparing with other part-based motion retrieval methods**
>
> Thanks for your advice. We have studied the previous works for 3D motion-text retrieval, and take the following works that are relevant to body parts as discussions. We distinguish our method and novelty from the following aspects:
>
> - First, in terms of the paradigm, we argue that our method explores the part motion modeling under a new paradigm of generation-augmented retrieval (GAR), which is different from [1] and [2]. This enables the testing augmentation option for performance boosting.
> - For detailed methodology, [1] considers the word-level features with a gating mechanism, to align with fine-grained motion features. However, it fails to model separate part motions on the motion encoder side with the coupled encoding of different parts. Meanwhile, it is difficult to learn this fine-grained motion-text correspondence due to the lack of explicit decoupling of linguistic knowledge as guidance. Besides, [2] explicitly considers the part motion description generation as extra knowledge. However, a naive solution for part alignment is adopted, i.e., directly using a contrastive learning objective. This ignores the difference of part motions compared with full-body motions, which are more coupled with each other due to the concurrence of different part motion patterns. In contrast, we apply LLM-based knowledge decomposition and propose part mixture decorrelation learning with directional relation regularization design, presenting a comprehensive part motion modeling framework.
> - Finally, we compare the performance of the above works in the following table, where we can see that our method has a desirable performance advantage.
>
> Table 2: Comparison of motion-text retrieval on HumanML3D dataset.
> | Method | M-to-T   |          |          |          |          | T-to-M    |          |          |          |          |
> |--------|--------------------------------|----------|----------|----------|----------|--------------------------------|----------|----------|----------|----------|
> |        | R@1                            | R@3      | R@5      | R@10     | MedR     | R@1                            | R@3      | R@5      | R@10     | MedR     |
> | [1]    | 11.36                          | 19.69    | 25.75    | 36.02    | 23.25    | 7.14                           | 17.4     | 24.02    | 34.67    | 24.00    |
> | [2]    | 9.29                           | 20.61    | 28.29    | 40.25    | 18.00    | 8.13                           | 19.69    | 27.07    | 39.18    | 18.00    |
> | **SGAR**   | **12.32**                          | **23.36**    | **30.47**    | **41.45**    | **16.00**    | **8.71**                           | **20.79**    | **27.92**    | **40.94**    | **17.00**    |
> | **SGAR++** | **11.76**                          | **23.47**    | **30.56**    | **43.18**    | **15.00**    | **9.38**                           | **22.23**    | **30.08**    | **42.58**    | **15.00**    |
>
> **The codes of [1,2] are not available. Therefore, we follow their test protocol that ignores the misjudgments caused by mirror augmentation for evaluation correction, and report our performance under this setting for fairness.*
>
> [1] Hierarchical Semantics Alignment for 3D Human Motion Retrieval, SIGIR 2024.
> [2] KinMo: Kinematic-aware Human Motion Understanding and Generation, arXiv 2024.
>
> ---
>
> **W3 & Q1: How to ensure LLM generated query is correct?**
>
> Thanks for the insightful question. This is an important problem that we have made efforts to study the corresponding design. We explain our consideration from the following aspects:
> - First, the generation of part motion texts based on the given full-body motion texts can be viewed as a process of knowledge decomposition or refinement from coarse to fine. From the perspective of users, the provided queries should include all the content that is intended to be emphasized. Otherwise, any other unrestricted associated content should be regarded as reasonable. Therefore, at this point, the LLM only needs to perform relatively simple part-wise decoupling of the provided coupled full-body motion texts. Many recent studies [1-5] have demonstrated that LLMs possess such capabilities, even in tasks that are more complex than human motion text data, which provides support for our design.
> - Technically, we carefully design the LLM prompts, i.e., a few-shot strategy with [udf] indicator for uncertainty, to improve the quality of generation.
> - On the other hand, due to the current limitations of LLMs and the existence of hallucination problems, this automated method cannot achieve 100% correct output results. Therefore, we randomly selected 200 samples and conducted manual checks, with an error rate of approximately 9%. Fortunately, our method has demonstrated a certain degree of robustness to the quality of part texts, as shown in Table 10 in the Appendix, which enables this design to bring expected performance improvement to motion-language models.
>
> [1] Large language models are versatile decomposers: Decomposing evidence and questions for table-based reasoning, SIGIR 2023
> [2] Least-to-Most Prompting Enables Complex Reasoning in Large Language Models, ICLR 2023
> [3] Mldt: Multi-level decomposition for complex long-horizon robotic task planning with open-source large language model, arXiv 2024
> [4] Reasoning Abilities of Large Language Models: In-Depth Analysis on the Abstraction and Reasoning Corpus, ACM TIST 2025
> [5] Tree of Thoughts: Deliberate Problem Solving with Large Language Models, NeurIPS 2023
>
> ---
>
> **Q2: Explain why testing is better without LLM**
>
> This can be explained from three aspects:
> - The key motivation is to decompose and model the part motion consistency to benefit the general motion-text alignment learning. Taking "walks and waves hand" as an example, with single global alignment learning, the model may incorrectly correspond the motion pattern of arm swinging to the verb "walk" rather than "wave hand", especially when training with a limited data scale. This means the difficulty of motion-text retrieval arises not only from the cross-modal alignment but also from the reorganization and decomposition of semantic concepts within the same modality. In this context, learning the correspondence of local concepts as primitives can lead to more precise, generalized local alignment, and ultimately enhance global understanding.
> - It is also beneficial to introduce more diverse part texts generated by LLMs, which alleviates the over-fitting problem with more data.
> - Moreover, as shown in the ablation study Table 2, the proposed part mixture learning and directional relation regularization further boost the performance with no LLM during testing. These designs introduce additional knowledge, e.g., novel mixed motion patterns and relational knowledge, to boost the overall performance and generalization.
>
> ---
>
> **Q3: Efficiency on the motion-to-text retrieval with LLM**
>
> Thanks for this valuable question! Actually, in motion-to-text retrieval, numerous inferences of LLM can degrade the real-time efficiency. Fortunately, we can alleviate this problem through some simple strategies. For example, the following two strategies can be considered
> - **Pre-computed Text Augmentation**: During the database construction stage, LLM can be used in advance to generate part motion descriptions as supplementary (or the database can use LLM to generate it offline during idle time). The augmented part texts are vectorized with the index established. When querying, we can directly retrieve the pre-augmented database to avoid the overhead of real-time generation.
> - **Hierarchical Query**: Another solution is to use hierarchical query. For instance, we can first use only the full-body motion texts to retrieve the Top-k candidates in the database, and then perform more fine-grained LLM augmentation on these Top-k texts with the GAR strategy. This can effectively reduce the number of LLM inference. To support this scheme, we provide the results of the hierarchical query when using different k (Top-k candidates). We can see that a small k (10~20) can also boost the performance significantly.
>
> Table 3: Results using hierarchical query with different k on motion-to-text retrieval on HumanML3D.
> | Top-k    | R@1   | R@3   | R@5   | R@10  | MedR  |
> |-------------|-------|-------|-------|-------|-------|
> | 0 (no LLM)  | 13.82 | 23.38 | 30.09 | 41.83 | 16.00 |
> | All test data (4380)  | 14.64 | 25.52 | 32.55 | 44.50 | 14.00 |
> | 20          | 14.65 | 25.60 | 32.59 | 44.39 | 14.00 |
> | 10          | 14.67 | 25.63 | 32.43 | 41.85 | 16.00 |

---

### Note · Authors · 2025-08-15

Dear Reviewers and AC,

Thank you and all reviewers for your valuable time and insightful feedback. We deeply appreciate the positive feedback on our work, such as
- Reviewer 5qdg: "Well-designed ablation studies", "novel part-mixture learning and directional relation alignment", "showing significant improvements over previous state-of-the-art methods".
- Reviewer JXyx: "reasonable LLM augmentation design", "novel and interesting part-mixture learning and directional relation alignment strategy", "provides promising results".
- Reviewer pgZA: "a novel framework for 3D human motion-text retrieval", "effectively designed components".
- Reviewer MCsr: "comprehensive experiments and competitive performance", "interesting part-mixture learning and directional relation alignment strategy", "ablation studies verifying the key designs".


Regarding the Reviewer 5qdg, JXyx and MCsr, we provided further explanation on our method design and motivation, with extensive experiment results for comparison and reproduction, demonstrating the improvement over the SOTA methods in terms of the retrieval performance and model generalization capacity to other downstream tasks.


For the concerns raised by reviewer pgZA, we have provided clarification on the method design and symbolic (W1,2,5), emphasized our text encoder design to defend the fairness of the experiment (W3), and supplemented extensive experiments to demonstrate the superior generalization ability of our method compared with the SOTA methods (W4).


After rebuttal, we are pleased that the concerns of the Reviewer JXyx and MCsr are mostly addressed according to their feedback sticking to positive ratings. Besides, with a direct mandatory acknowledgement and no further comments, Reviewer 5qdg and pgZA also do not present new issues during the discussion. We acknowledge all reviewers’ valuable suggestions and these revisions would be merged into the final version.


In summary, we have addressed the raised concerns with solid experiments and rigorous arguments. Our work introduces a novel generation-augmented retrieval paradigm, exploring the structural knowledge with effective alignment learning strategies. We demonstrate our significant effectiveness compared with the latest methods, including ChronRet (ECCV 2024), LaMP (ICLR 2025) and Lex (ICLR 2025). We kindly request your consideration on our extensive efforts and provided evidence in your decision.


Thank you again for your time and consideration.

---

### Decision · Program_Chairs · 2025-09-17

**Decision:**

Accept (poster)

**Comment:**

This paper introduces SGAR, a framework for 3D human motion–text retrieval that leverages LLM-based motion decomposition, part-mixture decorrelation learning, and directional relation alignment. The method achieves consistent improvements over prior state-of-the-art approaches (e.g., ChronRet, LaMP, Lex) and demonstrates generalization to related tasks such as cross-dataset retrieval and text-guided motion generation.

Across four reviewers, the paper is viewed positively: reviewers highlighted the novelty of the part-mixture and alignment strategies, the comprehensive ablation studies, and the strong empirical results. Concerns raised include: (i) limited discussion of computational efficiency, (ii) the need for clearer positioning relative to prior part-based and LLM-augmented motion retrieval works (e.g., CoMo, ECCV’24), (iii) missing comparisons with some recent baselines in applications such as motion generation, and (iv) reproducibility issues (lack of demos or multiple-seed results). The authors provided clarifications, additional experiments, and committed to addressing these points in the final version.

For the camera-ready, the authors should: (1) expand related work discussion to better position the novelty of LLM-based augmentation and part-level alignment, (2) provide runtime and resource usage details, (3) add or clarify comparisons with recent strong baselines in retrieval and motion generation, and (4) strengthen reproducibility by releasing code and, if possible, demos or averaged results.